



# Forecasting of regional methane from coal mine emissions in the Upper Silesian Coal Basin using the on-line nested global regional chemistry climate model MECO(n)(MESSy v2.53)

Anna-Leah Nickl[1], Mariano Mertens[1], Anke Roiger[1], Andreas Fix[1], Axel Amediek[1], Alina Fiehn[1], Christoph Gerbig[2], Michal Galkowski[2,3], Astrid Kerkweg[4,*], Theresa Klausner[1], Maximilian Eckl[1], and Patrick Jöckel[1]

[1]Deutsches Zentrum für Luft- und Raumfahrt, Institut für Physik der Atmosphäre, Oberpfaffenhofen, Germany
[2]Max Planck Institute for Biogeochemistry, Jena, Germany
[3]AGH University of Science and Technology, Krakow, Poland
[4]Institute of Geosciences and Meteorology, University of Bonn, Germany
[*]now at: Research Center Juelich, Institute of Energy and Climate, Juelich, Germany

**Correspondence:** Anna-Leah Nickl (anna-leah.nickl@dlr.de)

**Abstract.** Methane is the second most important greenhouse gas in terms of anthropogenic radiative forcing. Since pre-industrial times, the globally averaged dry mole fraction of methane in the atmosphere has increased considerably. Emissions from coal mining are one of the primary anthropogenic methane sources. However, our knowledge about different sources and sinks of methane is still subject to great uncertainties. Comprehensive measuring campaigns, as well as reliable chemistry climate models, are required to fully understand the global methane budget and to further develop future climate mitigation strategies. The CoMet 1.0 campaign (May to June 2018) combined airborne in-situ, as well as passive and active remote sensing measurements to quantify the emissions from coal mining in the Upper Silesian Coal Basin (USCB, Poland). Roughly 502 kt of methane are emitted from the ventilation shafts per year. In order to help the campaigns flight planning, we performed 6-day forecasts using the on-line coupled, three times nested global and regional chemistry climate model MECO(n). We applied three nested COSMO/MESSy instances going down to a spatial resolution of 2.8 km over the USCB. The nested global/regional model system allows for the separation of local emission contributions from fluctuations in the background methane. Here we introduce the forecast setup and assess the model skill by comparing different observations with the individual forecast simulations. Results show that MECO(3) is able to simulate the observed methane plumes and the large scale patterns (including vertically integrated values) reasonably well. Furthermore we receive reasonable forecast results up to forecast day four.


# 1 Introduction

In terms of radiative forcing methane is the second most important anthropogenically altered greenhouse gas (Myhre et al., 2013). The globally averaged dry mole fraction of methane has increased rapidly since 2007 (Nisbet et al., 2014, 2016), and its growth has even accelerated in 2014 (Nisbet et al., 2019; Fletcher and Schaefer, 2019), where the annual rise was 12.7

± 0.5 ppb. (Nisbet et al., 2019). The reason for the rapid methane growth in the atmosphere is currently under debate and discussed in several studies (Schaefer et al., 2016; Nisbet et al., 2016, 2019; Saunois et al., 2017; Thompson et al., 2018). The largest increase of methane is observed in the tropics and midlatitudes (Nisbet et al., 2019). A depletion in global $\delta^{13}$C indicates a shift from fossil fuel emissions towards more microbial sources (Schaefer et al., 2016; Nisbet et al., 2016, 2019). Nisbet et al. (2016) suggest natural emissions from wetlands as a result of positive climate feedback are the primary source

of the methane enhancement. On the contrary, Schaefer et al. (2016) propose that the increase in atmospheric methane since 2007 mainly originates from enhanced agricultural activity. Additionally, a change of the atmospheric oxidation capacity, i.e. a reduction of the OH sink, could play a role and may explain the shift in isotopic signature (Rigby et al., 2017). Increasing fossil fuel emissions could also explain the rise in atmospheric methane (Thompson et al., 2018). Shale gas is more depleted in $\delta^{13}$C relative to conventional gas and could be associated with the observed global depeltion in $\delta^{13}$C, too (Howarth, 2019). And

Schwietzke et al. (2016) pointed out that the fossil fuel emissions are 20 % to 60 % higher than previously thought. However, we still do not fully understand all factors that affect the sources and sinks of methane (Saunois et al., 2016). Furthermore, a reduction of anthropogenic emissions is attractive and inexpensive and due to its relatively short lifetime ($\sim$ 9 years), it could rapidly cause a change in the global methane budget (Dlugokencky et al., 2011). Comprehensive measurements and the use of chemistry climate models can therefore help to improve further climate change projections, and to develop potential climate

change strategies.

The AIRSPACE project (Aircraft remote sensing of greenhouse gases with combined passive and active instruments) aims for a better understanding of sources and sinks of the two most important anthropogenic greenhouse gases: carbon dioxide and methane. Several measurement campaigns within the project e.g. CoMet (Carbon Dioxide and Methane Mission), are carried out to increase the number of airborne and ground-based measurements of $CO_2$ and $CH_4$. CoMet 0.5 in August 2017

combined ground based in-situ and passive remote sensing measurements in the Upper Silesian Coal Basin (USCB) in Poland, where large amounts of methane are emitted due to hard coal mining (roughly 502 kt $CH_4$/yr, CoMet internal $CH_4$ and $CO_2$ emissions over Silesia, version 2 (2018-11), further denoted as CoMet ED v2). CoMet 1.0, that took place in May and June 2018, additionally included airborne in-situ as well as passive and active remote sensing measurements in the Upper Silesia and Central Europe. In order to localize the methane plumes and to obtain the best measurement strategies for the campaigns,

it is helpful to have reliable forecasts of the methane distribution in the atmosphere. We performed model based forecasts over the entire period of the campaigns using a coupled global and regional chemistry climate model. While local features are often not resolved in global climate models, it is important for the CoMet forecasts to resolve the local methane emissions from the coal mining ventilation shafts in the USCB. Therefore a smaller scale atmospheric chemistry model is required, which is provided by the on-line coupled model system "MESSyfied ECHAM and COSMO models nested n times" (MECO(n),



Kerkweg and Jöckel, 2012b; Mertens et al., 2016). To increase the resolution of our forecasts, we apply a nesting approach with three simultaneously running COSMO/MESSy instances down to a spatial resolution of 2.8 km. Section 2.2 describes the model setup and the implementation of two different methane tracers. We describe the details of the new forecast system (Sect. 2.3) and discuss its evaluation. We evaluated the model performance by comparing the methane mixing ratios simulated

by the two finest resolved COSMO/MESSy instances with airborne observational data. In Sect. 3 we show the comparisons with data that were sampled using three different measuring methods during the CoMet 1.0 campaign. Moreover, we assess the forecast performance firstly by internal comparison of the individual forecast days with the analysis simulations of CoMet 0.5 and CoMet 1.0 (Sect. 4.1) and secondly by comparison of the forecast results with the observations of CoMet 1.0 (Sect. 4.2).

## 2    Model and Forecast System

### 2.1    Model Description

The numerical global chemistry climate model ECHAM/MESSy (EMAC, Jöckel et al., 2010) consists of the Modular Earth Submodel System (MESSy) that is coupled to the general circulation model ECHAM5 (Roeckner et al., 2006). EMAC comprises various submodels that describe different tropospheric and middle atmospheric processes. It is operated with a 90 layer vertical resolution up to about 80 km altitude, a T42 spectral resolution (T42L90MA) and a time step length of 720 s. For our

purpose, EMAC is nudged by Newtonian relaxation of temperature, vorticity, divergence and the logarithm of surface pressure towards the ECMWF operational forecast or analysis data. Sea surface temperature (SST) and sea ice coverage (SIC), that are also derived from the ECMWF data sets, are prescribed as boundary conditions. The EMAC model is used as a global driver model for the coarsest COSMO/MESSy instance.

The model COSMO/MESSy consists of a Modular Earth Submodel System (MESSy, Jöckel et al., 2005) which is con-

nected to the regional weather prediction and climate model of the Consortium for Small Scale Modelling (COSMO-CLM further denoted as COSMO, Rockel et al., 2008). The COSMO-CLM is the community model of the German regional climate research community jointly further developed by the CLM-Community. Details on how the MESSy infrastructure is connected to the COSMO model are given in the first part of four MECO(n) publications (Kerkweg and Jöckel, 2012a). Several COSMO/MESSy instances can be nested on-line into each other in order to reach a regional refinement. For chemistry-climate

applications the exchange between the driving model and the respective COSMO/MESSy instances at its boundaries must occur with high frequency. This is important to achieve consistency between the meteorological situation and the tracer distribution. Furthermore, the chemical processes should be as consistent as possible. In MECO(n) the model instances are coupled on-line to the respective coarser COSMO/MESSy instance. The coarsest COSMO/MESSy instance is then on-line coupled to EMAC. Contrary to the off-line coupling, the boundary and initial conditions are provided by direct exchange via computer memory

using the Multi-Model-Driver (MMD) library. This coupling technique is described in detail in Part 2 of the MECO(n) documentation series (Kerkweg and Jöckel, 2012b). The chemical processes are described in submodels, which are part of MESSy. These submodels do not depend on spatial resolution and can be used similarly in EMAC and all COSMO/MESSy instances. A





detailed evaluation of MECO(n) with respect to tropospheric chemistry is given in the fourth part of the MECO(n) publication series (Mertens et al., 2016). In the present study we use MECO(3) based on MESSy version 2.53.

The MESSy submodel S4D (Jöckel et al., 2010) on-line samples the model results along a specific track of a moving object, such as air planes or ships. The simulation data is horizontally (and optionally also vertically) interpolated to the track and

sampled at every time step of the model. This guarantees the highest possible output frequency (each model time step) of respective vertical curtains along the track. The submodel SCOUT (Jöckel et al. 2010) on-line samples the model results as a vertical column at a fixed horizontal position. The high frequency model output is useful for comparison with stationary observations, such as ground-based spectroscopy or lidar measurements.

## 2.2 Model Setup

To resolve the local emissions from the ventilation shafts in the USCB, we operated MECO(n) with three nested instances, MECO(3), see Figure 1. The first COSMO/MESSy instance (hereafter called CM50) covers the European area and is operated at a resolution of 0.44° (~50 km) and with a time step length of 240 s. CM50 is on-line coupled to EMAC, resulting in a direct exchange of boundary conditions between the global model and the regional COSMO/MESSy model.

The second COSMO/MESSy instance (hereafter called CM7) covers the area over Central Europe and is operated with a res-

olution of 0.0625° (~7 km) and a time step length of 60 s. The smallest instance (hereafter called CM2.8) covers the Upper Silesia in Poland and thus also the target region. CM2.8 has a resolution of 0.025° (~2.8 km) and a time step length of 30 s. The individual finer COSMO/MESSy instances (CM7 and CM2.8) are on-line driven from the respectively coarser model domain (CM50 and CM7). In doing so, the respective coarser domain provides the boundary data for the smaller domain at each of its model time steps (EMAC: 720 s → CM50: 240s → CM7: 60s → CM2.8). Figure 2 shows an overview of the initial and

boundary data exchange between the different domains. CM50 and CM7 are operated with 40 vertical layers, and the smallest domain CM2.8 is operated with 50 vertical layers, that cover the atmosphere from the surface up to an altitude of 22 km. A sponge zone begins at 11 km which reaches the models top and nudges the models prognostic variables with increasing weights towards the driving model.

### 2.2.1 Methane tracers

CoMet aims to quantify the methane emissions in the USCB region, which actually arise from coal mining. In order to separate these emissions within our model, we defined two different methane tracers. One tracer only considers the point source emissions of the ventilation shafts (hereafter called PCH4) and the other tracer takes into account all methane emission fluxes

(hereafter called CH4_FX). Figure 3 shows an overview of both tracers, the involved submodels and the corresponding emission inventories. We initialized these two independent tracers for EMAC and for all three COSMO/MESSy instances equally. The initial conditions for the forecast simulations are derived from a continuous analysis simulation, which is described in detail in Sect. 2.3.



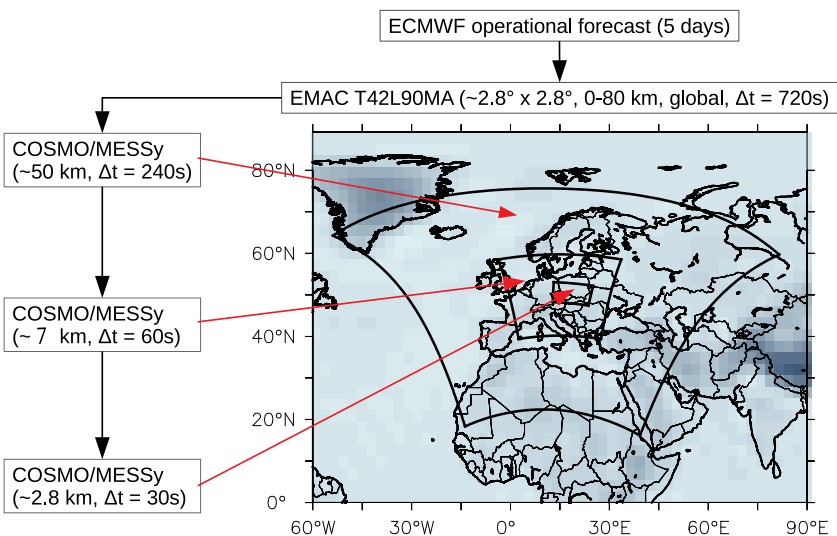

**Figure 1.** Overview of all three COSMO/MESSy domains over Europe (CM50), over Central Europe (CM7) and over the USCB in Poland (CM2.8), as well as the corresponding temporal and spatial resolution. The black arrows indicate the data exchange between the different models. The driving model EMAC is nudged towards divergency, vorticity, temperature and the logarithm of surface pressure from ECMWF. SST and SIC are prescribed as boundary conditions.

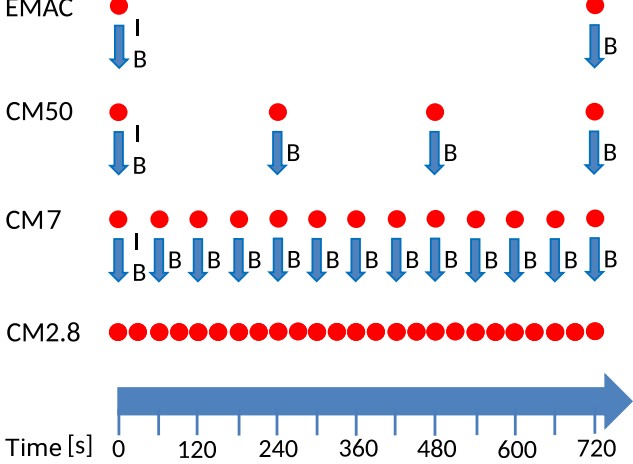

**Figure 2.** The illustration shows the initial and boundary data exchange between EMAC and the different COSMO/MESSy instances. The blue arrows symbolize the data exchange between the different model instances. B stands for boundary data and I for initial data. The red circles visualize the specific time steps of data exchange.





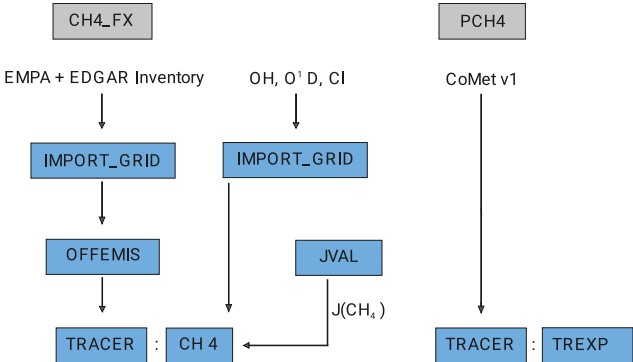

**Figure 3.** Illustration of submodels which are used for the different methane tracers. CH4_FX tracer (left side): Methane emission data and the oxidation reaction partners OH, O$^1$D and Cl are read from the netcdf files and transformed to the computational grid by the submodel IMPORT_GRID. OFFEMIS converts the emission fluxes into tracer tendencies, and the CH4 submodel simulates the chemical loss of methane using the predefined fields of the oxidation partners and the calculated photolysis rate from the submodel JVAL. PCH4 (right side): submodel TREXP is used for the point source emissions and tracer definition.

**CH$_4$ Point sources (PCH4)**

The PCH4 tracer considers only point source emissions, that are emitted by the ventilation shafts of the various coal mines in the USCB. In Fig. 4 the entire territory (49.90°N - 50.40°N latitude and 18.30°E - 19.40°E longitude) together with the location of the ventilation shafts are shown. To prescribe the emissions coming from the different shafts, we used CoMet

ED v1, an internal inventory mainly based on the European Pollutant Release and Transfer Register (E-PRTR 2014, Retrieved from http://prtr.eea.europa.eu, Feb 08, 2017), but also on data from Wyzszy Urzad Gorniczy 2014 (Retrieved from http://www.wug.gov.pl/download/5710.pdf, Feb 08, 2017). Further details on the names and exact position of the different mines can be found in the Supplement. The total point source methane emissions in this area are estimated to be 465 kt/a (CoMet ED v1 inventory). Emissions of single coal mines are split equally between the corresponding ventilation shafts. For

the definition of point sources, we applied the MESSy submodel TREXP that is described in detail by Jöckel et al. (2010).

**Gridded methane emissions (CH4_FX)**

The second tracer is called CH4_FX and includes all methane emission fluxes, anthropogenic and natural. We used an inventory which consists of two different parts, both monthly averaged: the year 2012 of the EMPA inventory (Frank, 2018) with a 1.0° × 1.0° grid resolution and the EDGAR v4.2FT2010 (Retrieved from http://edgar.jrc.ec.europa.eu, May 30, 2017) inventory

with a finer grid resolution of 0.1° × 0.1°. All anthropogenic (including rice cultivation) emissions are used from the EDGAR v4.2FT2010. Natural emissions and emissions caused by biomass burning are used from the EMPA inventory. The emission data are imported and transformed to the computational grid (IMPORT_GRID, Kerkweg and Jöckel, 2015). The emission fluxes are then converted into tendencies of the tracer CH4_FX (OFFEMIS, Kerkweg and Jöckel, 2012b, therein described



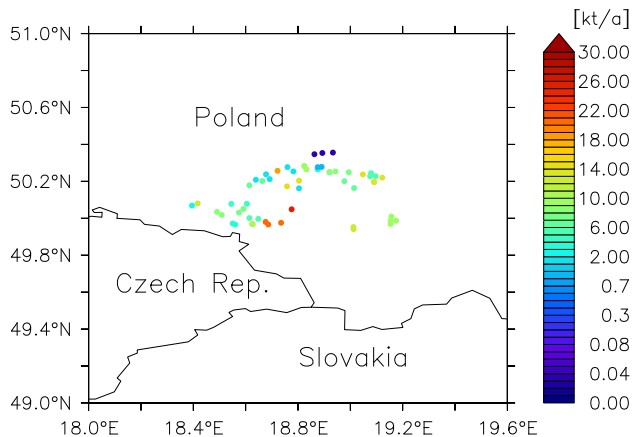

**Figure 4.** The map shows the locations and the emissions of methane in tons per year of the ventilation shafts in the USCB (CoMet ED v1). All ventilation shafts are gathered in the south-west of Poland close to the polish city Katowice and the Czech border.

as OFFLEM). Processes that are related to the methane chemistry in the model are described in the MESSy submodel CH4 (Frank, 2018). The submodel simulates the chemical loss of methane including the depletion by photolysis rate calculated by the submodel JVAL (Sander et al., 2014). The CH4 submodel uses predefined fields of the oxidation reaction partners OH, $O^1D$ and Cl which, for our setup, were derived as monthly averages (2007-2016) from a previous simulation and read by

IMPORT_GRID.

### 2.3  The Forecast System

In order to achieve the best initial conditions of PCH4 and CH4_FX, the daily forecast simulations are branched from a continuous analysis simulation. In the analysis simulation EMAC is nudged by Newtonian relaxation of temperature, vorticity, divergence and the logarithm of surface pressure towards the 6-hourly ECMWF operational analysis data. SST and SIC, derived

from the same data set, are prescribed as boundary conditions for EMAC. The initial conditions of CH4_FX have been derived as monthly climatological average (2007-2016) of the simulation SC1SD-base-01, which is similar to the RC1SD-base-10 simulation (Jöckel et al., 2016), however for the RCP8.5 emission scenario. PCH4 is initialized with zero. The starting dates of the analysis simulations are August 1st, 2017 for CoMet 0.5 and April 1st, 2018 for CoMet 1.0. This results in a spin up time of 8 days and 45 days, respectively. For the interpolation in time, starting and continuing the analysis simulation requires two

nudging time steps ahead of the simulated time. An analysis simulation which should start at 00:00 UTC, hence requires the nudging data of the time steps 06:00 UTC and 12:00 UTC. Once the respective time period is simulated and the corresponding restart file is written, a new forecast simulation is triggered. The forecast branches as a restart from the analysis simulation and simulates a time period of six days by using the 6-hourly ECMWF operational forecast data for the EMAC nudging. PCH4



and CH4_FX are automatically initialized from the restart files. Throughout this process the analysis simulation continues. The forecast system is visualized schematically in Fig. 5. As soon as the pre-processed nudging files become available, the analysis simulation runs for about 50 min. Each forecast simulation takes about 8 hours and the post processing takes another 1.5 to 2 hours. The 8 hours are for 144 message passing interface (MPI) tasks on an Intel Xeon E5-2680v3 based Linux Cluster (6

nodes à 12 dual cores), whereby 6, 18, 56, and 64 tasks were used for the model instances EMAC CM50, CM7, and CM2.8, respectively. In our example, a forecast that simulates a time period starting at forecast day one at 00:00 UTC, is readily post-processed on forecast day two at around 04:30 UTC (after approximately 28.5 hours). Throughout both campaigns, forecasts were delivered every 12 hours and made available online on a web page. In order to guarantee a continuous and uninterrupted supply of forecasts, we run the simulations alternately on two independent HPC (High Performing Computing) Clusters. An

example of a forecast web product, which shows the forecast starting on June 07, 2019 at 00:00 UTC can be found here: https://doi.org/10.5281/zenodo.3518926 (Jöckel et al., 2019). The post-processing included the vertical integration of PCH4 and CH4_FX into a total column dry air average mixing ratio, called $XPCH_4$ and $XCH_4$ for PCH4 and CH4_FX, respectively. It is calculated as follows:

$$XCH_4 = \frac{\sum(\chi_{CH_4} \cdot m_{dry})}{\sum m_{dry}} \tag{1}$$

where $\chi_{CH_4}$ is the methane mixing ratio, $m_{dry}$ stands for the mass of dry air in a grid box and summation is carried out over all vertical levels. Figure 6 shows the design of $XPCH_4$ and $XCH_4$ which appeared on the forecast website. It is an example of a snapshot during CoMet 1.0 simulated with CM2.8.

## 3   Evaluation of Analysis Simulation

### 3.1   Observational Data

During CoMet 1.0, methane was measured by active remote sensing. The instrument is an "integrated path differential absorption" (IPDA) Lidar called CHARM-F (Amediek et al., 2017), which was installed on board of the German Research Aircraft HALO (High Altitude and LOng Range). CHARM-F is operated by the German Aerospace Center (DLR) in Oberpfaffenhofen

and measures the weighted atmospheric columns of the methane dry-air mixing ratio from the surface to the flight altitude of the research aircraft. We compare our model results to the observations of the HALO D-ADLR flights on June 6th, 2018 and on June 7th, 2018. For simplicity, both data sets are hereafter called C1 and C2. See also Table 1, which lists all flights considered in this study and their abbreviations. Both data sets have a temporal resolution of 1 s and are already smoothed horizontally with a box window corresponding to 2 km flight distance.

Additionally, methane was sampled in-situ by Cavity Ring Down Spectroscopy (CRDS). A JIG instrument (Jena Instrument for Greenhouse Gases, Filges et al., 2015), that also measures the methane mixing ratio in-situ by CRDS, was installed on board of HALO and operated by the Max-Plank Institute for Biogeochemistry in Jena. Our model results are compared to the



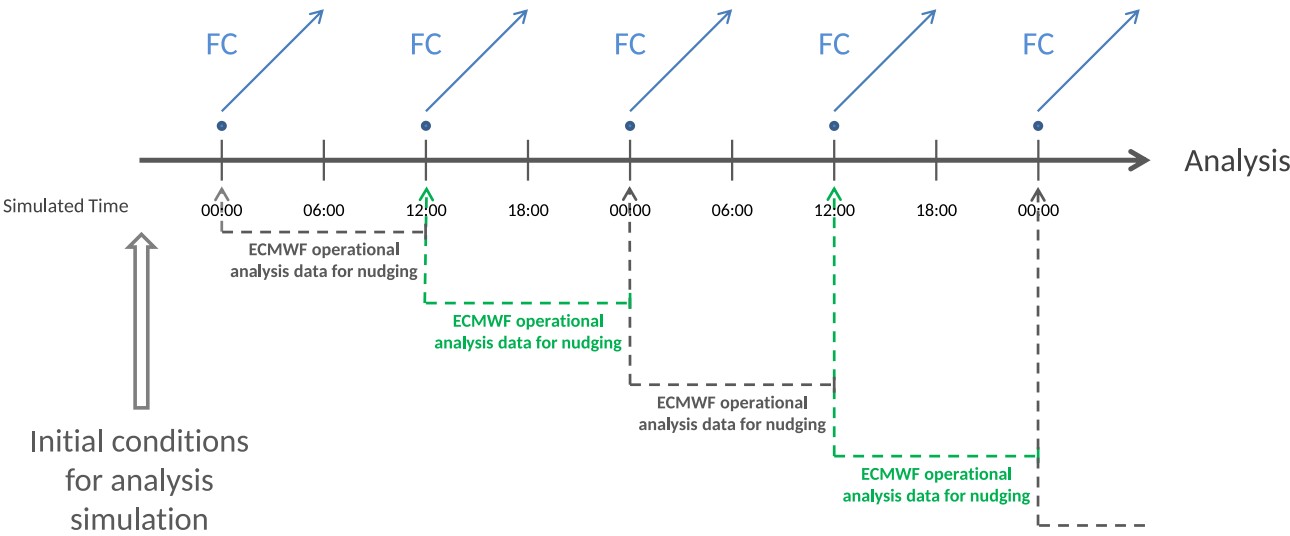

**Figure 5.** Chronology of the analysis simulation (dark grey) and the branching of the forecasts (FC, blue). The analysis simulation continues as soon as the ECMWF operational analysis data (two time steps ahead) are available for nudging. The required nudging time steps are indicated by the dotted lines in grey and green. A forecast simulation is branched (blue dots) every 12 hours from the analysis at 00:00 or 12:00 and simulates a time period of 6 days. The initial conditions are provided by restart files of the analysis simulation.

observations of the HALO flights on June 6th, 2018 and on June 7th, 2018. Data sets are abbreviated with J1 or J2 (see Table 1). Both data sets have a temporal resolution of 1 s. A Picarro CRDS G1301-m instrument was installed on board of the DLR research aircraft Cessna 208B (D-FDLR) and operated by the DLR in Oberpfaffenhofen. We compare seven flight observations to our model. Data sets are named accordingly P1 - P7 (see Table 1) and have a temporal resolution of 1 s.

5      Upon completion of CoMet 1.0, we conducted the analysis and forecast simulations again and used the specific geographical flight track coordinates (in degrees), pressure altitudes (in hPa) and time steps (in UTC) of all flights for the S4D submodel. The simulated data were then sampled as track-following curtains at each model time step; i.e. every 720 s, 240 s, 60 s and 30 s for EMAC, CM50, CM7 and CM2.8, respectively. However, our evaluation in this study only considers the two finest COSMO/MESSy instances CM7 and CM2.8. As the observed data has a finer temporal resolution than the model output, they

10    were averaged over 60 s for CM7 and over 30 s for CM2.8. In order to compare our model results with those of the CHARM-F measurements, we calculated the dry air mixing ratio between surface and aircraft (in the following referred to as $X_{fl}CH4$) using the S4D submodel output.





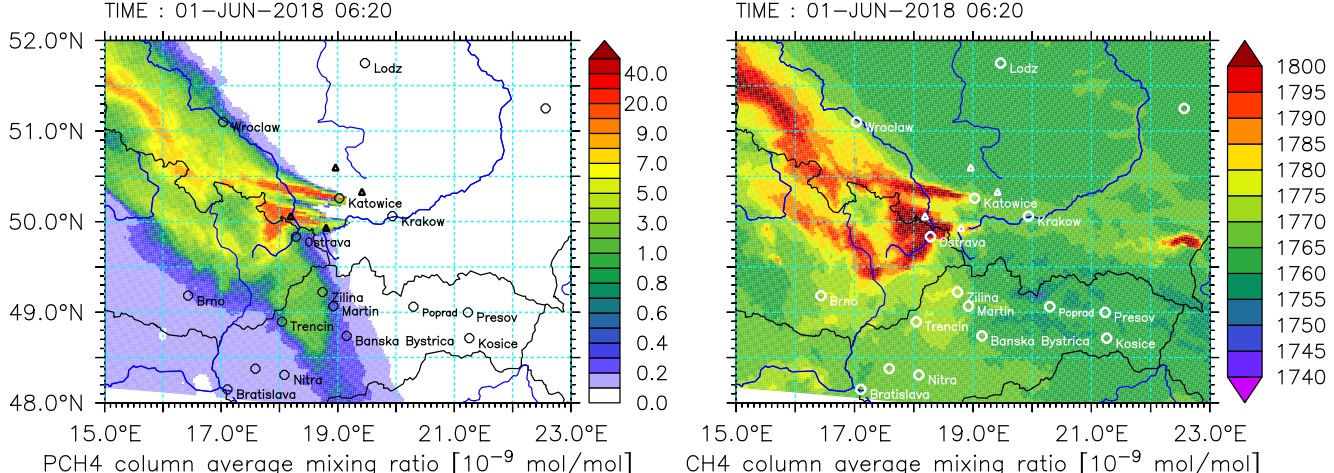

**Figure 6.** Snapshot of the methane forecasts during CoMet 1.0 simulated with the finest resolved COSMO/MESSy instance CM2.8. The total column dry air average mixing ratio in mol/mol is calculated for PCH4 (left) and CH4_FX (right). The area encompasses the USCB and shows the evolution of methane plumes in the atmosphere. Note that the color bar on the left is pseudo logarithmic for better visualization.

**Table 1.** Overview of the abbreviations for all observational methane data sets.

| Abbreviation | Flight | Instrument | Type of observation |
|:---:|:---:|:---:|:---:|
| C1 | HALO, 6th of June 2018 | CHARM-F | $XCH_4$ |
| C2 | HALO, 7th of June 2018 | CHARM-F | $XCH_4$ |
| J1 | HALO, 6th of June 2018 | JIG (with Picarro CRDS G2401-m) | in-situ |
| J2 | HALO, 7th of June 2018 | JIG (with Picarro CRDS G2401-m) | in-situ |
| P1 | D-FDLR, 29th of May 2018 | Picarro CRDS G1301-m | in-situ |
| P2 | D-FDLR, 1st of June 2018 | Picarro CRDS G1301-m | in-situ |
| P3 | D-FDLR, 5th of June 2018 | Picarro CRDS G1301-m | in-situ |
| P4 | D-FDLR, 6th of June 2018, morning | Picarro CRDS G1301-m | in-situ |
| P5 | D-FDLR, 6th of June 2018, afternoon | Picarro CRDS G1301-m | in-situ |
| P6 | D-FDLR, 7th of June 2018 | Picarro CRDS G1301-m | in-situ |
| P7 | D-FDLR, 11th of June 2018 | Picarro CRDS G1301-m | in-situ |

## 3.2 Comparison with Analysis results

As the analysis simulation is already nudged towards the ECMWF operational analysis data, we assume to reproduce the best possible meteorology. Thus, in order to find the best estimate of our model performance, the observations are compared to the analysis simulation results first. The model performance is analyzed with respect to pattern similarity and amplitude, i.e. root





**Table 2.** Summary of the results of the statistical analysis of C1 and C2 compared to the the simulated $X_{fl}CH4$. Listed are the root mean square error (RMSE) in $\mu$mol/mol and the normalized mean bias error (NMBE) in %. for the model domains CM7 and CM2.8.

| Flight | RMSE (CM7) | RMSE (CM2.8) | NMBE (CM7) | NMBE (CM2.8) |
|--------|------------|--------------|------------|--------------|
| C1 | 0.08 | 0.08 | -4.1 | -4.2 |
| C2 | 0.10 | 0.10 | -5.3 | -5.3 |

mean square error (RMSE), standard deviation, correlation coefficient and normalized mean bias error (NMBE):

$$NMBE = \frac{\sum(\chi_{sim} - \chi_{obs})}{n \cdot \overline{\chi}_{obs}} \cdot 100 \qquad (2)$$

where is $\chi_{sim}$ is the simulated methane mixing ratio, $\chi_{obs}$ stands for the observed methane mixing ratio and the summation is over all $n$ time steps. The results are presented in Sect. 3.2.1 (CHARM-F) and 3.2.2 (D-FDLR and HALO in-situ). In Sect.

3.2.3 we discuss all statistical results graphically.

### 3.2.1   Comparison with CHARM-F observations

Figure 7 shows the observed XCH$_4$ values of C1 and C2 as black lines. Red and blue dots display the simulated $X_{fl}CH_4$ of CM7 and CM2.8, respectively and all methane mixing ratios are given in $\mu$mol/mol. On both days the observed patterns agree well with the simulated patterns. Peaks in the observed methane mixing ratios are represented in CM7 as well as in CM2.8.

From a visual point, amplitudes also appear to be similar. Mismatches can be seen on 6 June, around 09:30 UTC (see Fig. 7 (a)) where model results are slightly shifted in time. Observed XCH$_4$ values follow a negative trend until 10:10 UTC, which is not simulated by the models. On June, 7 the observed amplitudes are larger than those of the model results. On both days CM7 and CM2.8 do not differ significantly from each other. However, small scale patterns are better resolved in CM2.8 and amplitudes slightly exceed those of CM7. Furthermore, the comparisons reveal a continuous and constant bias. Simulated $X_{fl}CH_4$ values

are shifted towards smaller values compared to C1 and C2. Table 2 lists the root mean square error (RMSE) in $\mu$mol/mol and NMBE in % for all comparisons of the observations with the model results. NMBE is negative for all cases and ranges from -4.1 % to -5.3 %. NMBE as well as RSME are lower for C1 (0.08 $\mu$mol/mol) as for C2 (0.10 $\mu$mol/mol), which confirm the assumption of higher mean amplitude similarity (standard deviation) on June, 6 as on June, 7.

### 3.2.2   Comparison with in-situ measurements

Here, we discuss the D-FDLR flights P4, P5 and P2. The remaining observations and their comparison to the model results are shown in the Supplement. Figure 8 shows the comparison of the methane in-situ measurements derived with the Picarro CRDS on board of D-FDLR. The results shown are for the two flights P4 and P5. Both measurement flights aimed to sample the emissions of all methane sources within the USCB. The flight routes surround the USCB and follow a back-and-forth pattern along a horizontal track downwind of the mines, crossing the methane plume several times at different heights (see Supplement

for details on the flight pattern). Figure 8 (a) and (b) compare the simulated CH4_FX tracer mixing ratios along the flight tracks





## (a)

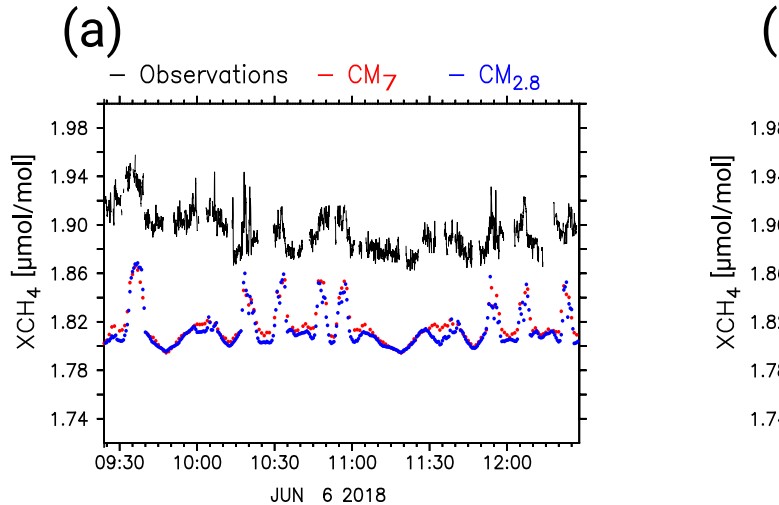

## (b)

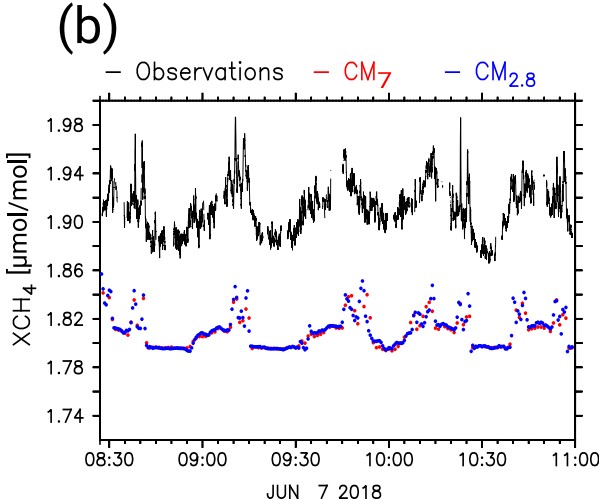

**Figure 7.** Results of the CHARM-F measurements and the S4D submodel sampling which was vertically integrated to yield $X_{fl}CH4$. $XCH_4$ of C1 (a) and C2 (b) are displayed in black and the simulated $X_{fl}CH_4$ of CM7 and CM2.8 are shown as red and blue dots, respectively. Mixing ratios are shown in $\mu$mol/mol. The time axes display the time of the specific flight in UTC.

to the observations. Pattern similarity is good for both flights and background methane shows little variability, but results of CM7 and CM2.8 are equally biased towards lower mixing ratios. Table 3 lists respective RMSE in $\mu$mol/mol and the NMBE in % for the comparison to both model instances with P4 and P5. On June, 6 in the morning, the NMBE is -4.4 % for CM7 and -3.6 % for CM2.8. Despite the negative bias, peak mixing ratios of CM7 and CM2.8 reach values close to those of the

observations, and around 10:15 UTC CM2.8 mixing ratios even exceed those of the observations. Although, generally in good agreement, CM7 and CM2.8 differ from each other from 10:00 UTC to 10:30 UTC. Due to the finer resolution, CM2.8 better resolves the small fluctuations observed within the plume and shows larger methane peaks than CM7. Regarding the afternoon flight (P5) model results again represent the observations well, in terms of time and location of the peaks. At 13:30 UTC the measured mixing ratios display a sharp increase, which is not distinctly presented in the model results. In general, observed

peaks are lower than the observed peaks of the morning flight. This is also seen in the simulation results. CM7 and CM2.8 differ from each other in such a way that CM2.8 better resolves the finer structure of the observed mixing ratios. Again, in CM2.8 methane peaks approach the observational peaks, but not as significantly, as seen for P4. Figure 8 (c) and (d) show the comparison between the simulated PCH4 values along the flight track to P4 and P5. The black line shows the observed $CH_4$ mixing ratios in $\mu$mol/mol and the red and blue dots show the model results for CM7 and CM2.8, respectively. As PCH4 only

considers the point source emissions without any background nor other methane source emissions, one can assume that the enhancements seen in the model and in the measurements originate from the ventilation shafts. Smaller variations within the background methane are consequently not present in the model results and stay at a constant level of zero. To allow a better comparison with the observations we added a constant offset of 1.85 $\mu$mol/mol in both plots. The simulated PCH4 mixing





**Table 3.** Summary of the results of the statistical analysis of P4 and P5 compared to the the simulated CH4_FX mixing ratios. Listed are the root mean square error (RMSE) in $\mu$mol/mol and the normalized mean bias error (NMBE) in %. for the model domains CM7 and CM2.8.

| Flight | RMSE (CM7) | RMSE (CM2.8) | NMBE (CM7) | NMBE (CM2.8) |
|--------|------------|--------------|------------|--------------|
| P4 | 0.10 | 0.10 | -4.2 | -3.6 |
| P5 | 0.09 | 0.10 | -4.3 | -4.3 |

ratios show a positive correlation with the major observed methane peaks. Although they do not show the same amplitudes, all methane elevations are simulated by the model. On June, 6 in the morning, CM2.8 values exceed CM7 values and clearly show a more distinct structure. In the afternoon this difference is even more remarkable. CM2.8 is able to simulate the variability very precisely, whereas CM7 does not resolve the smaller patterns seen in the observations (e.g. at 14:30 UTC).

Figure 9 (a) shows the comparison of the D-FDLR in situ observations of P2 at June, 1st 2018 to the CH4_FX mixing ratios simulated by CM7 and CM2.8. Again, a constant offset between observations and model results exists. Until 09:00 UTC atmospheric conditions were mostly stable and D-FDLR flew a back-and-forth pattern at a distance of 20 km downwind of the southwestern cluster of USCB mines. The very high observed mixing ratios around 08:22, 08:45 and 08:50 UTC result from the only slightly diluted plumes. Those enhancements (M1 and M2) are barely detectable in the S4D output. The M3

enhancement was sampled further to the north, downwind of the northern USCB mines. Panel (b) shows the corresponding simulated (CM2.8) methane profile along a flight track following curtain and the flight altitude in black. The model results in (b) show elevated methane mixing ratios at M1 and M2. The methane peaks are below the simulated boundary layer and below the flight track of P2. Consequently they are not visible in the S4D results (a). These findings indicate, that the simulated boundary layer during the morning is too low. Contrary, the observed methane peaks between 09:10 and 09:35 UTC (a) can be

seen in the S4D results. Here, the boundary layer already extended towards higher altitudes and the flight track is crossing the simulated methane plume (see panel (b)). Overall, CM7 and CM2.8 show smaller methane mixing ratios than observed.

Figure 10 shows the HALO in-situ measurements of methane J1 and J2 in black, along with the simulated CH4_FX of CM7 and CM2.8 in red and blue, respectively. All mixing ratios are displayed in $\mu$mol/mol. In an extra panel below the methane mixing ratios, atmospheric pressure along the flight track is plotted in hPa and indicates the changes in the flight altitude. The

pressure follows a steep up and down movement between 900 hPa and 200 hPa, which is because both flight paths were chosen to sample the vertical profile of methane in the atmosphere. The flight route of J1 and and J2 covered the USCB, but also parts which are lying outside the smallest model domain. Gaps in the simulated CM2.8 mixing ratios mark the temporal leaving of the smallest area.

Overall, observed and simulated methane mixing ratios correlate well with atmospheric pressure. Consequently methane cor-

relates negatively with flight altitude. Large scale patterns and amplitudes are very similar in both model instances as well as in the observations. Again, a constant bias towards smaller mixing ratios exists in the model results. It seems to be constant throughout the vertical levels. Table 4 lists RMSE and NMBE of J1 and J2 for the comparison with CM7 and CM2.8. NMBE have similar values ranging from -5.6 % to -5.9 %. Although in very good agreement, the model is not able to simulate the





**Figure 8.** D-FDLR in-situ sampled $CH_4$ mixing ratios (black lines) of P4 (left) and P5 (right), as well as the S4D submodel output at flight altitude for CM7 (red dots) and CM2.8 (blue dots). Panels (a) and (b) show the comparison with the CH4_FX tracer, whereas (c) and (d) shows the comparison with the PCH4 tracer. All mixing ratios are in $\mu$mol/mol. An offset of 1.85 $\mu$mol/mol is added to PCH4 for a better visualization. The time axis displays the time of the specific flight in UTC.

small scale fluctuations measured in the background methane at 400 hPa. Moreover, the model does not resolve the fine structure of the observations around 200 hPa. This can be seen in Fig. 10 (a) at 13:20 UTC and in Fig. 10 (b) at 12:00 UTC and 14:30 UTC. As the mixing ratio at this altitudes are strongly influenced by the boundary conditions of the global model, we would not expect that the model is able to reproduce these features. Contrary, methane variability at lower altitudes is well represented in CM7 and CM2.8. In general, CM7 and CM2.8 are in good agreement. RMSE for J1 is 0.12 $\mu$mol/mol for CM7 and 0.11 $\mu$mol/mol for CM2.8. For the comparison with J2, RMSE is similar with 0.11 $\mu$mol/mol for CM7 and CM2.8.



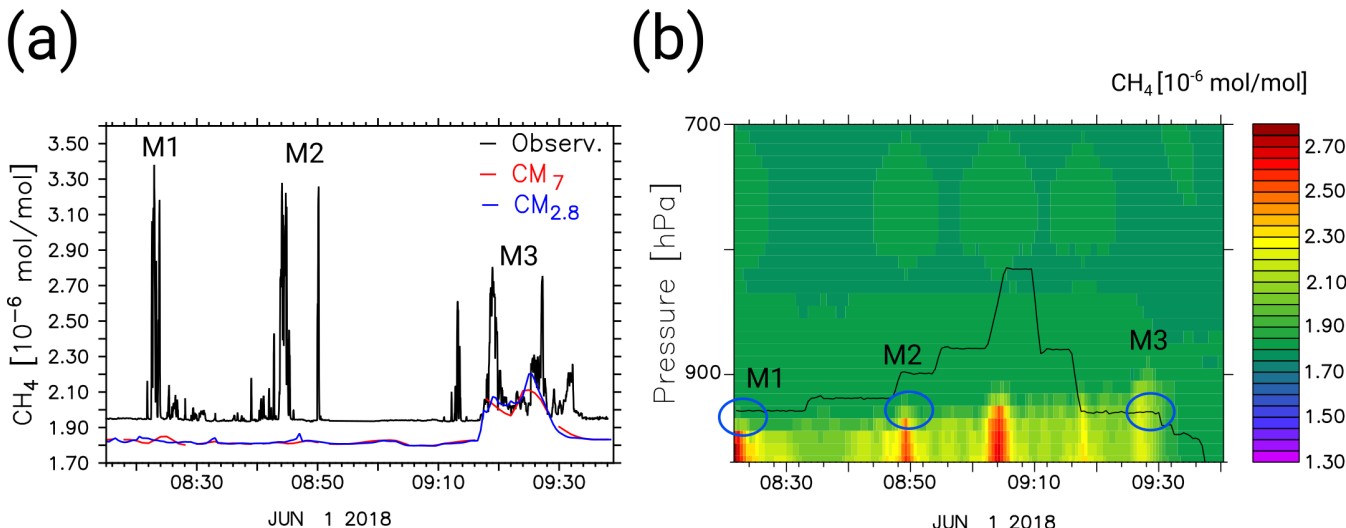

**Figure 9.** Panel (a) shows D-FDLR in-situ sampled $CH_4$ mixing ratios (black lines) of P2, as well as the S4D submodel output of CH4_FX at flight altitude for CM7 (red) and CM2.8 (blue). Panel (b) displays the corresponding flight altitude of P2 (black line) and the simulated profile (CM2.8) of the methane mixing ratio along this flight track. All mixing ratios are in $\mu$mol/mol. The time axes display the time of the flight in UTC. M1, M2 and M3 mark specific methane peaks seen in the observation (a) and in the simulated methane profile (b), but not necessarily below in the sampled S4D output at flight altitude (a).

**Table 4.** Summary of the results of the statistical analysis of J1 and J2 compared to the the simulated CH4_FX mixing ratios. Listed are the root mean square error (RMSE) in $\mu$mol/mol and the normalized mean bias error (NMBE) in %. for the model domains CM7 and CM2.8.

| Flight | RMSE (CM7) | RMSE (CM2.8) | NMBE (CM7) | NMBE (CM2.8) |
|--------|------------|--------------|------------|--------------|
| J1 | 0.12 | 0.11 | -5.9 | -5.7 |
| J2 | 0.11 | 0.11 | -5.7 | -5.6 |

### 3.2.3 Taylor Diagram

Taylor diagrams combine three statistical metrics to better compare and interpret different model performances. They summarize standard deviation (radial distance from the origin), correlation coefficient (angle) and centered RSME (dashed semi circles) in a single diagram (Taylor, 2001). Thanks to the normalization of standard deviation and centered RMSE (NRMSE), metrics become non-dimensional and different model results can be compared to each other. The point on the horizontal axis displaying a normalized standard deviation of 1 outlines the point where model results fit perfectly the observations. Figure 11 shows the results of the statistical analysis of CM7 (circles) and CM2.8 (triangles) compared to the observations (Table 1).

Although simulations visually suit well, the pattern and amplitude of C1 (see Fig. 7 (a)), correlation coefficient and centered NRMSE are rather low. Amplitudes and pattern statistically differ from the observations, which may be due to a temporal or





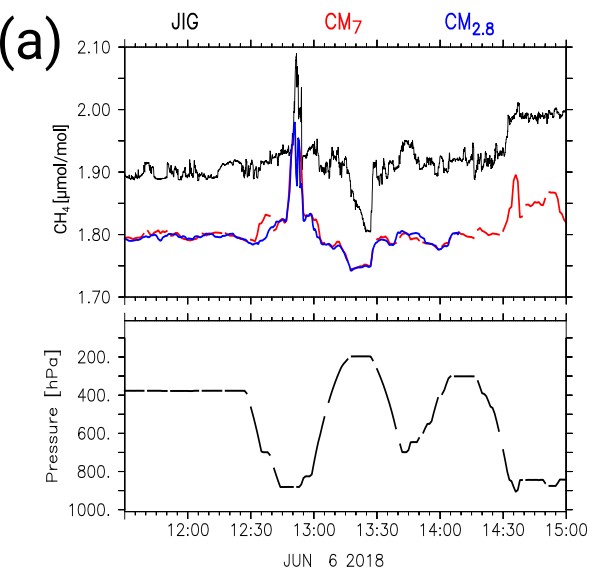

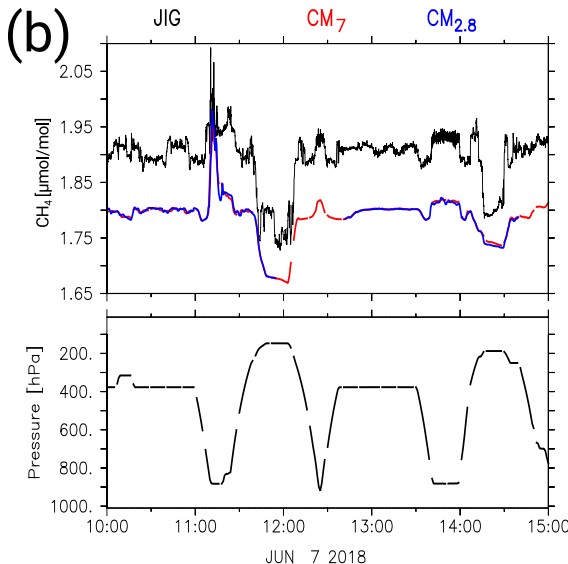

**Figure 10.** HALO in-situ sampled methane mixing ratios (black lines) of J1 (left) and J2 (right), as well as the S4D submodel output at flight level for CM7 (red dots) and CM2.8 (blue dots). All mixing ratios are in $\mu$mol/mol. Below the methane mixing ratios, the atmospheric pressure along the flight track is shown in hPa. The time axis displays the time of the specific flight in UTC.

spacial shift of the plume in the model at the beginning of June, 6. Contrary, the normalized standard deviations are almost the same. The standard deviation of C2 is larger than for C1, but here amplitude and pattern better fit to the observations. Overall CM7 and CM2.8 show similar performance. Model results corresponding to the smaller scale in-situ measurements P1, P4 and P5 have higher standard deviations than the observations. CM2.8 centered NRMSE are always larger than those of CM7. This

5    is also described in Sect. 3.2.2 where CM2.8, contrary to CM7, clearly exceeds the observed amplitudes. P3 and P6 are close to the reference line and P2 shows very low amplitudes compared to the observations, which is the result of very high methane mixing ratios M1 and M2 (see Sect 3.2.2) seen in the observations but not in the model results. Correlation coefficients do not show a specific pattern. Except for C1, all comparisons show lower correlation coefficients than J and C comparisons. P7 is not presented in the diagram as its normalized standard deviation is larger than 2.

10   The comparison to J1 and J2 is the best in this diagram. Results are closest to the reference point on the horizontal axis (normalized standard deviation = 1.0). Contrary to the comparisons with the P data sets, the CM2.8 standard deviation is closer to the observed standard deviation as the one of CM7. Correlation coefficients are very high, especially for CM7.



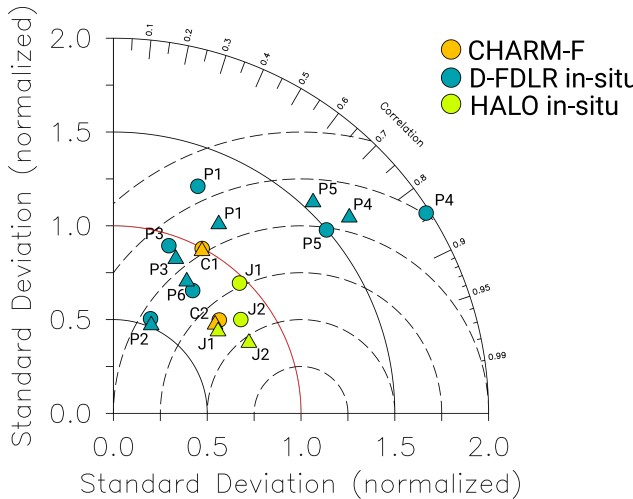

**Figure 11.** Taylor diagram summarizing normalized standard deviation (radius), correlation coefficient (angle) and centered NRMSE (dashed semi circles). Results show the comparison between observations and model domain CM7 (triangles) and CM2.8 (circles). Comparisons to C1 and C2 are displayed in orange, those to P1 until P6, in blue and those to J1 and J2 shown in green. P7 is outside the diagram.

## 4   Evaluation of Forecast Skill

A good forecast should be able to simulate both, amplitude and pattern variability, of the observed methane mixing ratios in the atmosphere. To identify the temporal evolution of the forecast skill with each forecast day, we therefore calculated skill scores (after Taylor, 2001) that consider standard deviation and correlation coefficient. We used the two different skill scores

$$S_{\mathrm{V}} = \frac{4(1+R)}{(\sigma_{\mathrm{f}} + 1/\sigma_{\mathrm{f}})^2 (1+R_0)} \tag{3}$$

and

$$S_{\mathrm{C}} = \frac{4(1+R)^4}{(\sigma_{\mathrm{f}} + 1/\sigma_{\mathrm{f}})^2 (1+R_0)^4} \tag{4}$$

which either emphasize the similarity of the amplitudes, or the similarity of the patterns. $R$ is the correlation coefficient between forecast and observation, $R_0$ is the maximum attainable correlation coefficient, and $\sigma_{\mathrm{f}}$ is the ratio of the standard deviation of the forecast to that of the observation. We assume $R_0$ to be 1, although in reality maximum correlation coefficients between observations and simulation can not be reached due to differences in spatial and temporal resolution. The skill ranges between 0 and 1, with small values indicating low skill and high values indicating high skill. We use the analysis simulation as a reference observation to evaluate a theoretical forecast skill. As the forecasts are branched from the analysis simulation we aim to quantify the deviation of the forecast from the analysis with increasing forecast day. The results are discussed in Sect. 4.1. In order to find the expected skill of the forecast, we further compare the different forecast days to the observations C1, C2, J1, J2, P4 and P5. Section 4.2 describes these results.





## 4.1 Theoretical Forecast Skill

We compared every single forecast day to the analysis simulation and calculated a daily skill score at each point on the respective two dimensional model grid. The skill was calculated for the simulated CH4_FX values. In order to compare CM7 and CM2.8, the analyzed area only covers the area obtained by removing the outermost 15 grid points of the CM2.8 domain

(relaxation area). Figure 12 assigns to each forecast day the average percentage of the area which reveals a skill score larger than 0.7. The results are shown in red for CM7 and in blue for CM2.8. Panels (a) and (b) refer to the different skill scores $S_V$ and $S_C$, respectively.

On forecast day I to III, CM7 shows slightly larger values than CM2.8. This is most terse for $S_C$, which puts greater emphasis on the correlation coefficient. However, differences between the two model instances are rather small. The forecast skill is

very large at forecast day I. Here, the forecasts are branched from the analysis simulation, which results in a good agreement between reference and forecast. Both skills decrease with increasing forecast day, whereby the skill in (b) shows a steeper decrease than the skill in (a). This suggests that the correlation between forecast and analysis reduces faster than the similarity of amplitudes. For $S_V$, the area which exceeds a threshold of 0.7, covers about 65 % and 40 % at forecast day II and III, whereas for $S_C$ it only covers about 50 % and 25 %, respectively. From forecast day IV onwards, less than 20 % ($S_V$) or 10 % ($S_C$) of

the area reveal a skill larger than 0.7.

Figure 13 shows the decline of the theoretical forecast skill within a Taylor plot by comparing the simulated $XCH_4\_FX$ values at a fixed location (Hotel Pustelnik in Poland, lat: 49.933024, lon: 18.799681, SCOUT output) to the analysis. 15 days of the CoMet 0.5 campaign are analyzed. Forecast day I (green circles) agrees best with the analysis. Most results gather around a point where the normalized standard deviation is almost 1, and the correlation between analysis and forecast is at

its maximum. Few exceptions show lower correlation coefficients down to 0.8 but stay close to the red reference line. The correlation coefficient on day II (purple triangles) ranges from about 0.33 to about 0.97 and the normalized standard deviation lies in the range between 0.5 to 1.5. On forecast days III (red diamonds) and IV (light blue stars), most normalized standard deviations cover the same range but the correlation coefficients decrease and do not show values larger than 0.89. Similar to what we observe in Fig. 12 the correlation coefficient seems to decrease faster or at least shows more variability, whereas the

standard deviation lies within a constant range. The lower correlation could be attributed to a displacement of the simulated plume in time or space, which would also explain the fact that the normalized standard deviation remains within the given range. Results for forecast days V (blue crosses) and VI (yellow circles) show correlation coefficients below 0.7 and a large variability in normalized standard deviation. Some points are outside the diagram and consequently not shown here.

## 4.2 Expected Forecast Skill

Figure 14 shows the skill score $S_V$ calculated for the different forecast days I to VI when compared to the observations C1, C2, J1 and J2 (see panel (a)) and to the observations P1 - P7 (see panel (b)). Results for CM7 and CM2.8 are shown on the left and right side, respectively. Contrary to the theoretical skill, where $S_V$ and $S_C$ clearly decrease with increasing forecast day, a



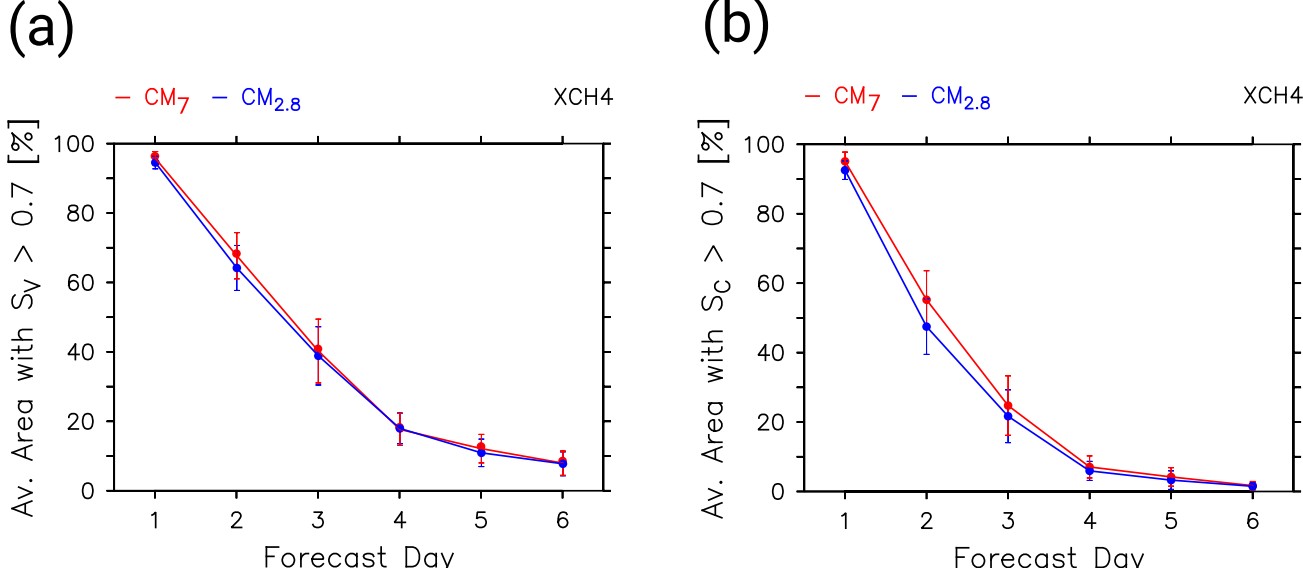

**Figure 12.** The graphic shows the evolution of the theoretical forecast skill with increasing forecast day for CM7 (red) and CM2.8 (blue). The vertical axis displays the average area (in %), according to the smallest model domain, with a Skill Score > 0.7. The horizontal axis shows the specific forecast days 1 to 6. The Skill Scores are calculated for each day at each grid point from $1^{st}$ to $22^{th}$ June 2018 (CoMet 1.0). It compares the CH4_FX total column mixing ratios of the forecast simulations to the CH4_FX total column mixing ratios of the analysis simulation. The error bars indicate the interval which contains 95% of all skill scores per day. $S_V$ (a) emphasizes variability and $S_C$ (b) emphasizes the correlation.

reduction of the skill is not obvious here. Considering the difference between the single observations, $S_V$ is highest for J1 and J2 with values above 0.8. They are followed by C1 and C2 with values above 0.65 (except for C1 on forecast day I) and P1, P4, P5 and P6 mainly showing a skill between 0.6 and 0.8. The skill is lowest for the comparison with P2 and P7. $S_V$ emphasizes the similarity of amplitude height between forecast and observation. This similarity seems to be highest with HALO in-situ

5    and CHARM-F observations. However, $S_V$ does not vary significantly among the different forecast days nor does it show any specific trend. Results for C1, P4, P5 and P7 drop at forecast day V, but increase again at forecast day VI. Contrary, P2 suddenly increases at forecast day V. Differences between CM7 and CM2.8 are rather small.

     Figure 15 summarizes the results of the skill score $S_C$. $S_C$ is generally lower than $S_V$, which is due to higher weighting of the correlation coefficient. Overall, skill is best for J1, J2, C2, P4 and P5, meaning that model and observations correlate well here.

10    P2 and P7 show again very low values (c. also Fig. 14). In panel (a), CM7 and CM2.8 show a similar pattern. The skill among the different forecast days almost stays at the same level or even increases until forecast day IV. Forecast day V and VI show lower skill, with lowest values for C1 at forecast day V. The skill for J1 and J2 shows generally lower values in CM2.8 than in CM7. In panel (b) the skill is highly variable among all forecast days until day IV. On forecast day V and VI, skill decreases for all comparisons, with very low values for P7.



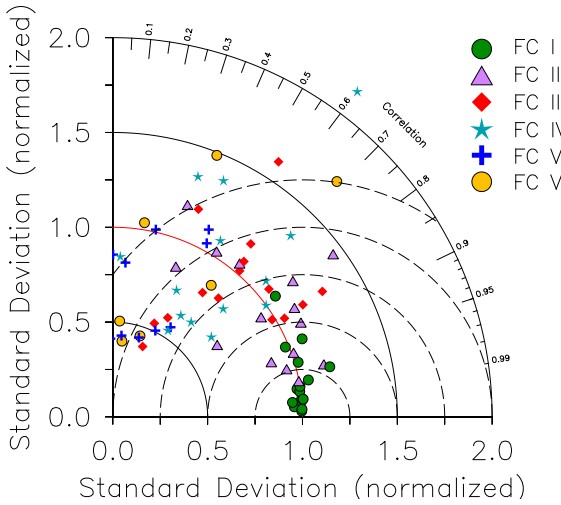

**Figure 13.** The Taylor diagram shows the comparison of the CH4_FX total column mixing ratios of the forecast simulations with the CH4_FX total column mixing ratios of the analysis simulation sampled with the SCOUT submodel at a selected location. Different symbols display the specific forecast days: Day 1 - green dots, day 2 - purple triangles, day 3 - red diamonds, day 4 - light blue stars, day 5 - blue crosses, day 6 - yellow circles. Correlation coefficient (angle), normalized standard deviation (radius) and normalized centered root mean square error (dashed semi circles) are calculated for each day from $10^{th}$ to $25^{th}$ August 2017.

Figure 16 shows the comparison of the individual forecast days to the observations: C1, C2 (triangles), J1, J2 (stars) or P4 and P5 (circles) in a Taylor diagram. Here we only show the results of the six best matches between MECO(n) forecasts and observations. The colors refer to the different forecast days II to VI and the inner dots mark the results that belong to the same sampling date (e.g. J1: star with dot, J2: star without dot). Forecast day I is not considered here, as it was not available for the

5  campaign (as described in Sect. 2.3). Panel (a) shows the results of the model domain CM7 and panel (b) shows the results of the smaller model domain CM2.8. In both cases, the comparisons with CHARM-F observations have low variability in standard deviation. All forecast days show a normalized standard deviation close to 1 (red reference line), meaning that all forecast days show similar amplitudes and match well the observations. While forecast day V (green triangles) shows the largest deviation from the reference, all other forecast days are closely grouped together and do not show a specific trend with increasing forecast

10  day. The comparisons with the HALO in-situ observations (stars) reveal similar results. The normalized standard deviations are close to 1 and do not show a clear pattern. The normalized standard deviations of the D-FDLR in-situ comparisons (circles) are again gathered, regardless of a particular forecast day, but show more internal variability (between the different forecast days) and higher deviations from the reference line. CM7 (a) results even show that the last forecast days stay closer to the reference line, which means that their amplitude height resembles the amplitudes of the observations better. Yet, this is however just a

15  result for two measurements sampled at the same day. The correlation coefficient varies between the different observations (data point with or without dots). And, in contrast to the normalized standard deviation, the correlation coefficient shows a





**Figure 14.** The bar plots show the calculated skill scores $S_V$ for the comparison of forecast day I to VI (horizontal axis) to the observations. The colors refer to the different observations. Panel (a) displays the results for C1, C2, J1, and J2. Panel (b) shows the results for P1-P7. CM7 and CM2.8 results are shown on the left and right side, respectively.

decreasing trend with increasing forecast day. Forecast day V and VI mostly show lower correlation coefficients than previous forecast days II, III and IV. Apart from a few exceptions the correlation of the forecast days III and IV are slightly higher than forecast day II. The decrease of correlation with increasing forecast day is most obvious when comparing with D-FDLR in-situ observation (i.e. for the observation targeting on small scale features). Deviations in correlation coefficient seem to be slightly

5    higher in CM7, compared to CM2.8



**Figure 15.** The bar plots show the calculated skill scores $S_C$ for the comparison of forecast day I to VI (horizontal axis) to the observations. The colors refer to the different observations. Panel (a) displays the results for C1, C2, J1, and J2. Panel (b) shows the results for P1-P7. CM7 and CM2.8 results are shown on the left and right side, respectively.

## 5 Discussion

Overall, the comparison of the analysis simulation with airborne derived measurements show that MECO(n) is able to sim-
5 ulate the observed methane plumes reasonably well. This the intended result given the fact, that EMAC is nudged towards



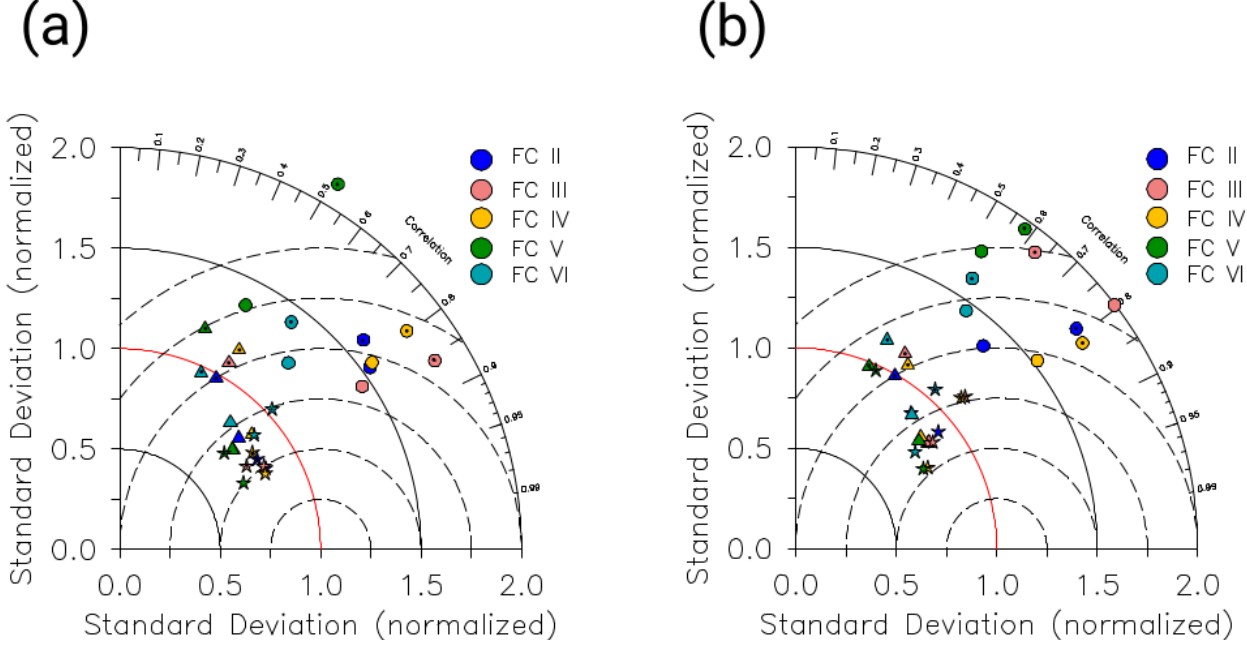

**Figure 16.** The Taylor plots summarize the comparison of the forecast simulations with the observations: CHARM-F (triangles), HALO in-situ (stars) and D-FDLR in-situ (circles). Different colors display the specific forecast days: Day 2 - blue, day 3 - orange, day 4 - yellow, day 5 - green, day 6 - light blue. Dots (C1, J1 and P4) or no dots inside shapes help to distinguish between the different flights. Correlation coefficient (angle), normalized standard deviation (radius) and normalized centered root mean square error (dashed semi circles) are calculated for CM7 (a) and CM2.8 (b).

the ECMWF data at a coarser resolution (T42 spectral truncation), and CM50, CM7 and CM2.8 are nested into each other and only driven by relaxation at their boundaries by the next coarser model instance. Nevertheless, a continuous and constant offset of the simulated CH4_FX to all observations results from all model instances. As the bias is constant at all altitudes, it is most likely not caused by shortcomings in the vertical transport in the model. Instead, global increase of methane emis-
5    sions (Nisbet et al., 2019) could explain the discrepancy between the observations and the model results that are based on EDGAR v4.2FT2010 for anthropogenic emissions and on the EMPA inventory (Frank, 2018) for all other emissions fluxes. Apart from that, the timing of the simulated peaks is in good agreement for all observations. Compared to CHARM-F observations, the simulated $X_{fl}CH_4$ peaks show similar or slightly lower amplitudes. The vertical methane gradient measured by the JIG instrument is well represented by MECO(n). Besides the smaller variations in the background methane, the model
10    results correlate well with the measured methane mixing ratios at different altitudes. By comparing the small scale D-FDLR in-situ measurements, the simulated amplitude of the peaks is mostly overestimated. This applies particularly for the CM2.8 results. Anthropogenic emissions in the EDGAR v4.2FT2010 inventory differ from the latest release EDGAR v4.3.2 (Retrieved from http://edgar.jrc.ec.europa.eu, Feb 04, 2019). Total anthropogenic methane emissions for the USCB (here: lon. 18.30°E - 19.40°E, lat. 49.90°N - 50.40°N) are 1636 kt/a in EDGAR v4.2FT2010 and only 605.6 kt/a in EDGAR v4.3.2. The over-





estimation of local methane plumes in the model can therefore be explained by the overestimation of methane emissions in EDGAR v.4.2FT2010. However, we also see high peaks in the methane mixing ratios of the D-FDLR in-situ observations, which are very low in the model results or not present at all. This is mainly the case when D-FDLR sampled very close to the ventilation shafts (see results for P3 in the Supplement). Here, the model can not resolve the very localized enhancements.

Larger grid sizes lead to instantaneous dilution of the simulated mixing ratios close to the ventilation shafts. Or, as seen for June 2nd 2018, the simulated boundary layer is too low and lies below the flight altitude. Consequently, the enhanced methane mixing ratios are not present in the S4D results along the flight track at flight altitude. Additionally, PCH4 which only considers the emissions of the ventilation shafts, is in good agreement with the CH4_FX tracer and the observed methane elevations. This indicates that the observed methane plumes actually originate from coal mining. Although the assessment of the single

point source emissions is not part of the current study, it should be noted that the different PCH4 point sources can be switched on and off in the model individually. This provides a good tool to distinguish between different sources and to assign them to the different measurements. When compared to the small scale D-FDLR in-situ measurements, PCH4 correlates well with the observed methane emissions. Contrary to the CH4_FX results, the point source tracer does not overestimate the emissions. However, it neither shows the same emission strength as the observations. PCH4 peaks have considerably lower amplitudes

than observed. The reason for this discrepancy is the different emission inventories for CH4_FX and PCH4. The sum of all methane emissions in CoMet ED v1 used for PCH4 is 465 kt/a, which is less than a third of the EDGAR v4.2FT2010 emissions (summed over the corresponding grid boxes). Updated estimates of emissions from CoMet ED v2 (based on E-PRTR 2016 (Retrieved from http://prtr.eea.europa.eu, Nov 07, 2018)) indicated larger emissions of 502 kt/a and some changes in the distribution of emissions, following structural and operational changes in the mining sector over the period between reporting

years (2014 and 2016 for CoMet DB v1 and v2, respectively). This implies that the simulated PCH4 using the latest emission inventory CoMet ED v2 are expected to match the observed amplitudes better. Another reason for the underestimation of the simulated PCH4 peaks might be the fact that we assume a temporally constant methane release from the ventilation shafts. But in reality the emitted amount of methane varies from day to day. This might probably have a small influence on the results, but would not explain the large differences between PCH4 and the observations. Overall, CM7 is able to simulated the large

scale observations (HALO in-situ) and the vertically integrated methane (CHARM-F) as precisely as CM2.8. When compared to small scale measurements (D-FDLR in-situ) the model overestimates the observed peaks. This is especially true for the finer resolved CM2.8, where methane mixing ratios are larger than the mixing ratios simulated by CM7. Smaller grid cells may catch locally enhanced methane mixing ratios in the plume, whereas coarser grid cells cover a larger portion of the methane plume and mixing ratios may be more diluted. Additionally, CM2.8 is able to better simulate the fine structure of the small

scale observations. However, the differences are rather small and the observed methane peaks are well represented in both model instances.

The theoretical forecast skill illustrates the deviation of the forecast from the analysis simulation. Results show a decreasing trend with increasing forecast day. Nevertheless, correlation and amplitude similarity of a single forecast days show a broad variation. The evaluation of the expected forecast skill reveals even less clear results. The amplitudes seem to be constant or at

least do not show any specific trend with increasing forecast day, but the correlation between observation and forecast slightly



decreases. Forecast day V seems to yield the lowest skill for P4 and P5 and also for C1 and J1, which is less obvious as for P4 and P5. Due to the fact that these observations were sampled during only one day, namely the 6th of June 2018 in the morning and in the afternoon, all comparisons for the fifth forecast day are related to the forecast simulation start date 2nd of June. Disagreement may be due specific meteorological situation of this day. In order to make a general statement about the forecast

skill, it would be necessary to compare additional observations within a broader time span.

## 6   Conclusions

We successfully conducted 6-day-forecast simulations of methane with the on-line coupled three times nested global and regional chemistry climate model system MECO(3). The forecasts branch from a continuous analysis simulation, where EMAC is nudged towards the operational ECMWF analysis data. This is essential for appropriate initial forecast conditions. We

continuously delivered the forecasts during CoMet 0.5 and 1.0 and analyzed the model and forecast performance with respect to the observations. The advantage of using the global/regional model system is, that we are able to simulate both, the point source emissions and the background methane. For the latter, it is essential to provide lateral boundary conditions to the nested model instances, which are consistent with the meteorology, i.e. the dynamical boundary conditions. This makes it possible to distinguish between local source emissions and fluctuations in the background methane, which is important for the

quantification of different methane sources. Even though the data for Newtonian relaxation are first coarsened to a horizontal grid resolution corresponding to the T42 spectral truncation, and then nested three times down to a spatial resolution of 2.8 km, MECO(3) is able to simulate the observed methane plumes correctly. Overall, the vertically integrated values, e.g. total column average mixing ratios, and the large scale patterns, such as the vertical gradient of methane, are well represented. However, limitations exist for the simulation of small scale patterns. A bias reduction as well as a better agreement of small

scale simulated methane amplitudes with the observations may be achieved by updating the applied emission inventories to the EDGAR v.4.3.2 inventory for anthropogenic emissions and the latest information on point source emissions (CoMet ED v2). Furthermore, we obtained decent results up to forecast day IV. The skill score calculated for all forecast day is reasonable. However, due to the limited number of comparable observations, the skill score might be not representative and its interpretation must be treated with caution. All observed methane peaks are well represented in both model domains CM2.8 and CM7. For

the purpose of the field campaign, it is therefore sufficient to perform the forecasts with CM7 only.

*Code availability.*   The Modular Earth Submodel System (MESSy) is continuously further developed and applied by a consortium of institutions. The usage of MESSy and access to the source code is licenced to all affiliates of institutions which are members of the MESSy Consortium. Institutions can become a member of the MESSy Consortium by signing the MESSy Memorandum of Understanding. More information can be found on the MESSy Consortium Website (http://www.messy-interface.org). We used MESSy v2.53

30       .



*Data availability.* The simulation results are available on request from the first author. The operational analysis and forecast data, used for nudging are licenced by the ECMWF. For further details, see https://www.ecmwf.int.

*Author contributions.* Astrid Kerkweg, Patrick Jöckel and Mariano Mertens developed the model system MECO(n) and Patrick Jöckel the forecast chain. Mariano Mertens set up the different model domains. Anna-Leah Nickl, Patrick Jöckel and Mariano Mertens planned and
carried out the forecast simulations. Theresa Klausner (with support of Anke Roiger and Andreas Fix) provided the emission inventory CoMet ED v1. Alina Fiehn and Michal Galkowski provided the emission inventory CoMet ED v2. Axel Amediek and Andreas Fix performed the CHARM-F measurements and Axel Amediek provided the processed data. Alina Fiehn, Maximilian Eckl and Anke Roiger carried out the D-FDLR in-situ measurements and Alina Fiehn provided the processed data. Michal Galkowski and Christoph Gerbig performed the HALO in-situ measurements and provided the data. Anna-Leah Nickl performed the simulations and analyzed the model results with input from
Patrick Jöckel and Mariano Mertens. Anna-Leah Nickl drafted the manuscript to which all authors contributed.

*Competing interests.* The author declare that they have no conflict of interest.

*Acknowledgements.* We gratefully acknowledge the funding of CoMet by the Deutsche Forschungsgemeinschaft (DFG, German Research Foundation) - Project Number 316646266 - SPP 1294, within the Priority Program "Atmospheric and Earth System Research with the "High Altitude and Long Range Research Aircraft (HALO)". We acknowledge the funding of the scientific data analysis by the Federal
Ministry for Research and Education (BMBF) within its AIRSPACE project through grants no. FKZ 01LK1701A and FKZ 01LK1701C. We acknowledge DLR VO-R for funding the young investigator research group "Greenhouse Gases". We thank Heidi Huntrieser (DLR-Institute of Atmospheric Physics, Oberpfaffenhofen, Germany) for the helpful comments on a previous version of the manuscript. We further gratefully acknowledge Jarek Necki and Justyna Swolkien from AGH University of Science and Technology, Krakow, Poland for providing the shaft locations for the internal databases of CoMet ED v1 and CoMet ED v2. We thank Winfried Beer (DLR-Institute of Atmospheric
Physics, Oberpfaffenhofen, Germany) for providing the java script of the forecast web page and the administration of the forecast web service. We acknowledge the computational resources provided by German Climate Computing Centre (DKRZ) in Hamburg.



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
