# Peer review of "Hind- and forecasting of regional methane from coal mine emissions in the Upper Silesian Coal Basin using the on-line nested global regional chemistry climate model MECO(n)(MESSy v2.53)"

_Geoscientific Model Development, 2019_

## Referee Comment (RC1) · Anonymous Referee #1 · 10 Dec 2019

General Comments: This paper provides an evaluation of a modeling forecast system setup to forecast methane plumes emitted from coal mines in the Upper Silesian Coal Basin in Poland. The aim is to forecast the methane plumes in order to assist with the flight planning of several measurement campaigns. An evaluation of the skill of the model at two different resolutions (2.8 and 7 km) as compared to three different airborne observational datasets is presented.

The authors present a comprehensive overview of the biases in this model evaluation paper which is of interest to the scientific community and fits within the scope of the

GMD journal. However, in general the paper is lacking an in-depth interpretation of the results. More could be discussed in terms of the sources of uncertainty and error. While the authors provide a reasonable interpretation of the results in the Discussion and Conclusion sections, the paper would be more interesting to read if more interpretation and analysis was given throughout the paper, rather than just reporting the biases. In terms of grammar, the paper is generally well written, but the author is inconsistent in using the past and present tense. The paper should be written consistently in either the past or present tense.

Specific Comments: Title and Abstract: I find the title and abstract misleading because most of the evaluation presented in this paper is focused on the analysis simulation rather than the forecast simulation. The title should be changed to something like "Modeling and forecasting of..." to reflect this. In addition, the abstract and introduction should also reflect that most of the evaluation is focused on 1) assessing the impact of the model's spatial resolution on the simulation of methane plumes originating from ventilation shafts in the coal mines, 2) assessing the uncertainty in the model's methane concentrations using different air-borne measurements, and 3) comparing the results of using two different emission inventories on peak methane concentrations over the coal mines.

1 Introduction: Page 2, Line 7: Explain isotope carbon-13 and how it can be used to infer sources of ch4 emission.

2 Model and Forecast System: The authors mention updating the applied emissions inventory to EDGARv4.3.2 which could help in reducing the biases. They could also consider using the CAMS-GLOB-ANT anthropogenic global emissions which are currently used by the ECMWF-IFS models and are based on the EDGARv4.3.2 and CEDS inventories and extrapolated to the current year (https://eccad3.sedoo.fr/).

2.2.1 Methane Tracers: Please provide an explanation as to why you are evaluating two tracers. Is it to compare the different emission inventories? If so, it should be clearly

stated. Otherwise, if it is just to get the other sources of methane emissions, and if the internal inventory of point sources is more accurate, why not replace the EDGAR emissions over the coals mines with these point sources? The authors should also consider using the CAMS-REG-AP regional inventory for Europe which is developed by TNO in the Netherlands and can be downloaded from the ECCAD data repository (https://eccad3.sedoo.fr/). These emissions are provided up to the year 2016 and are based on more detailed regional information than the global inventories.

Page 7, Line 12: Explain in more detail what is meant by ". . .however, for the RCP8.5 scenario. . .". Why is a scenario used?

3 Evaluation of Analysis Simulation: Please clearly explain exactly what is meant by "analysis simulation" for readers who are not familiar with forecast systems. It should be stated that the analysis simulation is constrained by the meteorology.

3.1 Observational data: The flight pattern for J1 and J2 should also be provided. Regarding the flight pattern shown in S2 for P4 and P5, it is redundant to show altitude on the y-axis and in the color-scale. Instead latitudinal information would be more useful.

3.2 Comparison with Analysis results: It is shown here that the model has a slight systematic negative bias in background CH4. Possible sources of the bias, which is presumably inherited from the global model, should be discussed (i.e. emissions, representation of chemical processes, etc.). Specifically, what impact could the prescribed OH field have on the simulated CH4 concentrations in terms of a chemical sink? Moreover, since the aim of the forecast system is to simulate methane plumes arising from the coal mines, I would suggest that the rest of the analysis be performed on the anomalies (with respect to background values) of the simulated and observed methane concentrations rather than the absolute values. This would remove the model's bias in the background methane concentrations and allow for a more straightforward comparison of the peaks related to emissions from the ventilation shafts of the coal mines.

Page 11, Line 13: It can't be seen on the graph that small scale patterns and are better

resolved in CM2.8. The sentence should be taken out or more proof provided.

3.2.2 Comparison with in-situ measurements: In discussing Figure 8(a) and (b), the authors state that "Despite the negative bias, peak mixing ratios of CM7 and CM2.8 reach values close to those of the observations...", however, it seems that since the simulated background methane has a negative bias (>10 umol/mol) then the model is actually overestimating the increase in the peak methane mixing ratios in the plumes compared to the observations, in particular for CM2.8. Could the authors comment or clarify this point? An evaluation of the anomalies would have been an effective way to remove the model's systematic bias from the background methane and evaluate the model's ability to reproduce the peaks observed in the methane plumes.

Can any conclusions be drawn regarding the relationship between the stability of the boundary layer, spatial resolution and the model's performance in simulating the methane plumes? Accurately simulating the PBL is critical in forecasting the methane plume. Has their model's PBL scheme been evaluated elsewhere? If so, it should be referenced here and discussed.

3.2.3 Taylor Diagram: I don't think that this section adds much information that hasn't already been presented in the timeseries plots. I would suggest to either summarize the results in a meaningful way or to remove it. For example, can you draw any conclusions about the model's bias with regard to the different types of observations? Why are the biases with the J observations lower than the P observations, and why do Tables 3 and 4 show the contrary? The authors should either present a full analysis of the differences in the biases (i.e. instrument type, PBL height, time of day, concentration in the plume, location, windspeed and direction, etc), or simply report on the range of uncertainty that is found using these three datasets which is already quite useful information in terms of assessing the model's skill.

4.1 Theoretical Forecast Skill: I'm not convinced that the Taylor diagram brings any additional information that can't be deduced from Figure 12.
4.2 Expected Skill Score: Please explain exactly what the expected skill score is because the fact that the model's skill does not decrease in the same manner as the theoretical skill score does not make sense to me. If we assume that the analysis is a "perfect simulation", and compared to the forecast simulation the theoretical forecast skill decreases to almost zero by day 6, how is it possible that the expected forecast skill in comparison to the observations is essentially the same on day 1 as day 6?

The authors present the model biases using different observations but more explanation or interpretation would be appreciated. For example, on page 19 line 1 it is stated that "...Sv is highest for J1 and J2..." but no explanation/speculation is offered as to why.

Again, I don't think that the Taylor diagram presented in Figure 16 adds any new information. It is clear from the plots in Figures 14 and 15, that the model's skill score for predicting the J observations is higher than for the other observations, especially the P observations. What would be interesting is for the authors to offer an explanation as to why this is the case. Why is there more variability in the skill score for the P observations than for the J or C observations? Unless the authors can draw some interesting conclusions such as this, I would suggest removing the Taylor diagrams and replacing them with the HALO and D-FDLR flight patterns.

Page 20, line 7: The authors state "All forecast days show a normalized standard deviation close to 1...meaning that all forecast days show similar amplitudes...". In theory, this can't be deduced from the standard deviation alone.

Technical Corrections: The author should go through the entire paper, especially (but not only) Section 3.1 and make sure they are consistent with either using the past or present tense.

Abstract: Line 4: change "measuring" to "measurement" Line 8: Change the sentence to read "In order to help with the flight planning during the campaigns..."

[Figure]

1 Introduction: page 2, Line 20: change "climate change strategies" to "climate change mitigation strategies"

2 Evaluation of Analysis Simulation: Page 12, line 10: change sentence to "...observed peaks in the afternoon flight are lower those of the morning flight."

Page 13, line 4: change "very precisely" to "more precisely"

Page 13, line 6: change "constant offset" to "systematic bias"

Page 13, line 12: change "...simulated boundary layer..." to "...simulated boundary layer height..."

5 Discussion: page 22, line 5: There is something wrong with the sentence "This the intended result given the fact, ...". Perhaps a word is missing.

---

## Referee Comment (RC2) · Anonymous Referee #2 · 6 Jan 2020

This paper describes the setup and application of a nested atmospheric transport modeling system to support an aircraft measurement campaign with daily model forecasts of methane over a coal-mining region in Poland. The performance of the forecasts is assessed both in terms of theoretical skill (comparing 1-6 day $CH_4$ forecasts with analyses) and in terms of "expected skill" (or actual skill) by comparison with aircraft observations of total column and in situ $CH_4$. The model is shown to be capable of simulating the structures and amplitudes of the $CH_4$ observations well, although this is more a qualitative than a quantitative statement since there is a large uncertainty in the

underly-ing emission inventories and since there is no comparison with other modeling systems.

The paper is well written, clearly structured, and the analyses are detailed (sometimes too detailed) and sound. MECO(n) is an impressively flexible model system capable of online nesting multiple in-stances of a regional model (COSMO) in a global model (ECHAM). The paper presents a relevant application of the model, which takes full advantage of its nesting and online processing capabili-ties (e.g. sampling the model fields at each time step along aircraft trajectories).

Supporting aircraft measurement campaigns requires models with sufficient resolution (much bet-ter than the horizontal and vertical distance travelled during the flights), but it is not clear a priori, what resolution is really needed and whether very high resolution brings sufficient added value to justify the additional computational cost. By comparing the results of two different model instances with a resolution of 7 km and 2.8 km, re-spectively, the paper shows that the results (CH4 sampled along the aircraft tracks) are very comparable and that the higher resolution does not bring a great benefit, though some small-scale details were better resolved. An interesting but also surprising finding is that the model skill (evaluated against observations) did not depend clearly on fore-cast lead-time, i.e. a 3- or 4-day forecast performed equally well as a 2-day forecast. Unfortunately, there is little discussion of this result. Overall, I consider the publication acceptable with minor revisions, but I have a few main points and a number of small corrections/suggestions.

Main points

- There are two methane tracers, PCH4 and CH4_FX, the first one representing emis-sions on-ly from coal mining and the second all emissions (anthropogenic+natural) plus background CH4. The anthropogenic emissions in CH4_FX are based on EDGAR v4.2FT2010. The authors need to check how large emissions in the USCB region are in EDGAR in comparison with the total emissions of the COMET ED v1 inventory

used for PCH4. To my understanding, fugitive emissions from solid fuels belong to category 1B1 (see IPCC 1996 reporting guide-lines), which is available as separate category in the EDGAR inventory. Without a compari-son of these numbers, it is diffi-cult to understand the results presented in Figure 8, which suggest and overestimation of the amplitude of CH4 enhancements for the tracer CH4_FX but an underestimation for the tracer PCH4. Furthermore, how do coal-mining emissions compare with other emissions e.g. from agriculture in this region (according to EDGAR?).

- The simulation results are biased low because of a too low background. This should not be surprising considering that the simulation was initialized from a monthly clima-tological av-erage of a period, when atmospheric CH4 was lower than in 2018. This bias is thus arbitrary and not of interest for the study (we are much more interested in the excursions from the background), but it dominates much of the statistics discussed and presented in the tables. I therefore suggest computing an overall bias (e.g. mean difference averaged over all flight sections measuring background) and subtract this constant offset from all simulation data, at least when computing the RMSE and NMBE statistics. In the current tables, the RMSE is of the order of 0.1 umol/mol, which is of a similar magnitude as the amplitude of the ob-served CH4 peaks, which would actually suggest a very poor model skill.

- The discussion on forecast skill is rather lengthy, especially the discussion of the Taylor dia-grams. I found it useful to summarize the results of all model-observation comparisons in a Taylor diagram as shown in Figure 11, but I am much less convinced of the use of Figures 13 and 16 summarizing the 1- to 6-day forecast skills at a single location (Fig. 13) and for the aircraft measurements (Fig. 16). Much of the information is already conveyed by the other figures. The discussion of Figures 13 and 16 is lengthy and not providing much addi-tional insight. Furthermore, one should be very careful in the interpretation of the results presented in Fig. 11, since much of the findings are simply a consequence of the different flight patterns. The high correlations in the HALO in situ measurements (J1, J2), for exam-ple, are primarily due to the large

altitude changes on these flights probing a large vertical gradient in CH4. But also horizontal flight patterns may critically affect the results, depend-ing on the complexity of the pattern, the overall distance flown, the time spent in sampling background versus polluted air, etc.

Minor points:

- Page 2, Line 28: Please explain what you mean by "internal"

- P4, L4: You mention that the simulation data can optionally be interpolated vertically. Was such vertical interpolation applied, or were the simulated fields only taken from the closest vertical layer?

- P6: Figure 3 could be improved. The black text in the blue boxes is a bit difficult to read.

- P7, L4: Were the O1D and Cl fields obtained from a full chemistry simulation?

- P7, Lines 13-18: I didn't really understand these sentences: Why do you need an "interpola-tion in time"? Why are data of time steps at 06:00 UTC and 12:00 UTC needed if the nudg-ing requires two time steps AHEAD of the simulated time, which starts at 00:00 UTC?

- P8, L9: I thing HPC stands for High Performance Computing (not "Performing").

- P9, Figure 5: Why are the grey and green dashed lines with arrows going from right to left? This seems to suggest that e.g. the forecast starting at 12:00 is branched from an analysis, which has seen nudging data between 12:00 and 24:00 on that same day. Is this really true?

- Figures 7 and 8: The flight patterns or at least the altitude profiles should also be shown in the main body of the paper, not just in the supplement, because this is es-sential infor-mation. It is important to know, for example, whether the individual peaks correspond to different plumes or whether the same plume was sampled back and forth

multiple times. It is also important to know whether changes in CH4 mole fractions are due to changes in flight altitude rather than due to transecting a plume.

- Tables 2, 3 and 4: I suggest adding the correlation coefficients (or R-square as a measure of the variance explained).

- P13, L21: I think it would be useful to show a vertical profile of CH4 for this flight to demonstrate that the model captures the vertical gradient of CH4 quite accurately.

- P17, equations of skill scores: Does one of these skill scores correspond to the dashed line in Figure 11? Is so, please mention.

- "Expected skill" doesn't sound right to me. What about "Actual skill", or "True skill"?

- P18, L1: For which period (how many days) did you compare the forecasts with the analysis simulation?

Corrections:

- Page 2, Line 28: "in the Upper Silesia" -> "in Upper Silesia"

- P4, L12: I suggest using "time step" instead of "time step length" here and in the following sentences.

- P13, L4: "very precisely". I would rather say "quite precisely"

- P13, L5: "at June" -> "on June"

- P13, L12: Change to "below the top of the boundary layer"

- P13, L24: "correlate well" has a positive connotation. "correlate closely" sounds better to me in this case.

- P14, L3: Change "Contrary," to either "In constrast," or to "On the contrary," here and at other places.

- P14, L4: "expect that the model is able" -> "expect the model to be able"

- P15, L8: "suit well" -> "fit well"

- P15, L9: Isn't the NRMSE high rather than low?

- P16, L1: "spacial" -> "spatial"

- P20, L14: "amplitude height" -> "amplitude"

- P22, L5: "This the intended result given" doesn't sound right.

- P24, L6: "the boundary layer is too low" -> "the boundary layer is too shallow". The top of the boundary layer can be too low, but not the boundary layer itself.

- P24, L13: "PCH4 correlates well with the observed methane emissions". There was no ob-servation of emissions but only of concentrations.

- P24, L23: "might probably" -> "might"

- P24, L22: "forecast day" -> "forecast days"

---

## Author Comment (AC1) · 26 Feb 2020

In black we repeat the referees comments, in red are our replies.

**Anonymous Referee #1**

General Comments: This paper provides an evaluation of a modeling forecast system setup to forecast methane plumes emitted from coal mines in the Upper Silesian Coal Basin in Poland. The aim is to forecast the methane plumes in order to assist with the flight planning of several measurement campaigns. An evaluation of the skill of the model at two different resolutions (2.8 and 7 km) as compared to three different airborne observational datasets is presented.

The authors present a comprehensive overview of the biases in this model evaluation paper which is of interest to the scientific community and fits within the scope of the GMD journal.

Dear Referee, thank you very much for the positive view of the overall study.

However, in general the paper is lacking an in-depth interpretation of the results. More could be discussed in terms of the sources of uncertainty and error. While the authors provide a reasonable interpretation of the results in the Discussion and Conclusion sections, the paper would be more interesting to read if more interpretation and analysis was given throughout the paper, rather than just reporting the biases.

We revised the manuscript according to your specific comments below.

In terms of grammar, the paper is generally well written, but the author is inconsistent in using the past and present tense. The paper should be written consistently in either the past or present tense.

We now use consistently past tense in the revised version.

Specific Comments: Title and Abstract: I find the title and abstract misleading because most of the evaluation presented in this paper is focused on the analysis simulation rather than the forecast simulation. The title should be changed to something like "Modeling and forecasting of. . ." to reflect this.

Thank your for this hint. We changed the title, using the word "hindcasting".

In addition, the abstract and introduction should also reflect that most of the evaluation is focused on 1) assessing the impact of the model's spatial resolution on the simulation of methane plumes originating from ventilation shafts in the coal mines, 2) assessing the uncertainty in the model's methane concentrations using different air-borne measurements,

We are grateful to this specific comment and changed the abstract accordingly.

and 3) comparing the results of using two different emission inventories on peak methane concentrations over the coal mines.

As the purpose of this study was not comparing two different emission inventories we did not include such a statement in the abstract. We rather provide an improved explanation of using the EMPA/EDGAR inventory and the point source inventory in the revised manuscript and below (comment 2.2.1 Methane tracer).

1 Introduction: Page 2, Line 7: Explain isotope carbon-13 and how it can be used to infer sources of ch4 emission.

We now give an explanation on stable carbon isotopes in the introduction part.

2 Model and Forecast System: The authors mention updating the applied emissions inventory to EDGARv4.3.2 which could help in reducing the biases. They could also consider using the CAMS-GLOB-ANT anthropogenic global emissions which are currently used by the ECMWF-IFS models and are based on the EDGARv4.3.2 and CEDS inventories and extrapolated to the current year (https://eccad3.sedoo.fr/).

We are grateful for this hint for our future studies, and we mention this emission inventory in the outlook of the revised manuscript.

2.2.1 Methane Tracers: Please provide an explanation as to why you are evaluating two tracers. Is it to compare the different emission inventories? If so, it should be clearly stated. Otherwise, if it is just to get the other sources of methane emissions, and if the internal inventory of point sources is more accurate, why not replace the EDGAR emissions over the coals mines with these point sources?

The purpose of using these two tracers was not to evaluate the different emission inventories. The EMPA/EDGAR inventory (used for the CH4_FX tracer) was used to provide a methane distribution, which contains all known methane sources (within the nested regions including the "background" methane which is advected into the model domain). The point source emission inventory (CoMet ED v1) was used **in addition** for a second methane tracer tracing only the emissions from the coal mine ventilation shafts. In this way, we are able to trace methane enhancements (of the first tracer, CH4_FX, which is equivalent to what has been measured) back to the coal mine emissions. In other words, the "point source tracer" was used as additional diagnostic for tracing the individual emission solely from the point sources (which have been the focus of the campaign). In that context, it was not our purpose to evaluate the point source emission inventory (at least not in the present study), and therefore we did not replace the point sources in the EMPA/EDGAR inventory.

We improved the corresponding explanation in the revised manuscript.

The authors should also consider using the CAMS-REG-AP regional inventory for Europe which is developed by TNO in the Netherlands and can be downloaded from the ECCAD data repository

(https://eccad3.sedoo.fr/). These emissions are provided up to the year 2016 and are based on more detailed regional information than the global inventories.

Thank you for pointing on the emission inventories, which we will consider in future simulations.
One issue, however, with regional emission inventories in our nested global/regional setup is that the regional inventories need to cover at least our nested region, and that the inventories used for the global and the regional domains must be consistent (e.g. w.r.t. potential biases), because otherwise artefacts at the regional model boundaries can occur. This has to be checked carefully.

Page 7, Line 12: Explain in more detail what is meant by ". . .however, for the RCP8.5 scenario. . .". Why is a scenario used?

This sentence refers to the simulation we use for the initialization.
The initial conditions of CH4_FX are derived as monthly climatological average (2007-2016) of the simulation SC1SD-base-01, which is similar to the RC1SD-base-10 simulation (described in detail by Jöckel et al., 2016). Whereas for RC1SD-base-10 the prescribed emissions of the last available year (2011) have been used for later years as well, SC1SD-base-01 has been performed with the boundary conditions of the RCP8.5 emission scenario. We used this for initialization as our current best guess including transient boundary conditions after 2010.
Since this is confusing, we removed the sentence in the revised manuscript.

3 Evaluation of Analysis Simulation: Please clearly explain exactly what is meant by "analysis simulation" for readers who are not familiar with forecast systems. It should be stated that the analysis simulation is constrained by the meteorology.

We guess that the referee refers to section 2.3 here, because here the forecast system is explained. We rephrased accordingly:
"In order to achieve the best initial conditions of PCH4 and CH4_FX, the daily forecast simulations are branched from a continuous analysis simulation, which is essentially a hind-cast simulation until the start of the forecast day."

Furthermore, in section 3, we rephrase the first sentence:
"As the analysis simulation is nudged towards the ECMWF operational analysis data, we assume that this simulation reproduces the observed meteorology best."

3.1 Observational data: The flight pattern for J1 and J2 should also be provided. Regarding the flight pattern shown in S2 for P4 and P5, it is redundant to show altitude on the y-axis and in the color-scale. Instead latitudinal information would be more useful.

Thank you for pointing this out.
We added Figure 9, showing the flight pattern for J1 (a) and J2 (b).
We additionally added Figure 11, showing the flight pattern for P4 (a) and P5 (b), and the two flight routes (c) and (d) to the corresponding section. The flight pattern show latitude, pressure and bias-corrected CH4_FX, the flight routes show longitude, latitude, pressure at flight level and the exact positions of the coal mines. We removed the Fig. S1 from the Supplement.

3.2 Comparison with Analysis results: It is shown here that the model has a slight systematic negative bias in background CH4. Possible sources of the bias, which is presumably inherited from the global model, should be discussed (i.e. emissions, representation of chemical processes, etc.). Specifically, what impact could the prescribed OH field have on the simulated CH4 concentrations in terms of a chemical sink?

Thank you for the comment. We now discuss possible sources of the bias at the beginning of the analysis.

Moreover, since the aim of the forecast system is to simulate methane plumes arising from the coal mines, I would suggest that the rest of the analysis be performed on the anomalies (with respect to background values) of the simulated and observed methane concentrations rather than the absolute values. This would remove the model's bias in the background methane concentrations and allow for a more straightforward comparison of the peaks related to emissions from the ventilation shafts of the coal mines.

We agree. We therefore define an average background value using the most frequently occurring difference between in-situ measurements and model results, subtract this bias and redo the statistical analyses. Unfortunately, we cannot apply the same bias to the integrated total column average mixing ratios (comparison to CHARM-F measurements). This is discussed in Section 3.2.

Page 11, Line 13: It can't be seen on the graph that small scale patterns and are better resolved in CM2.8. The sentence should be taken out or more proof provided.

Thank you for the hint. We removed this sentence.

3.2.2 Comparison with in-situ measurements: In discussing Figure 8(a) and (b), the authors state that "Despite the negative bias, peak mixing ratios of CM7 and CM2.8 reach values close to those of the observations. . .", however, it seems that since the simulated background methane has a negative bias (>10 umol/mol) then the model is actually overestimating the increase in the peak methane mixing ratios in the plumes compared to the observations, in particular for CM2.8. Could the authors comment or clarify this point? An evaluation of the anomalies would have been an effective way to remove the model's systematic bias from the background methane and evaluate the model's ability to reproduce the peaks observed in the methane plumes.

Thank you for this comment. Indeed, this sentence is misleading and is actually supposed to point on the overestimation of the model. We now changed the sentence accordingly. As already stated above, we also revised the analysis after bias correction.

Can any conclusions be drawn regarding the relationship between the stability of the boundary layer, spatial resolution and the model's performance in simulating the methane plumes? Accurately simulating the PBL is critical in forecasting the methane plume. Has their model's PBL scheme been evaluated elsewhere? If so, it should be referenced here and discussed.

To our knowledge, there is only one paper by Collaud et al. (2014) analyzing the COSMO simulated PBL height. However, they did not systematically study the impact of the model resolution. Moreover,

they come to a conclusion, which contradicts our finding (e.g. Fig. 12 in the revised manuscript, Fig. 9 in the original manuscript and corresponding discussion). We want to stress, however, that our simulations are short term and cannot provide a detailed analysis of the PBL height. Yet, these questions are highly interesting for further studies, but beyond the scope of our manuscript.

The revised manuscript includes a brief discussion on that in the discussion section.

3.2.3 Taylor Diagram: I don't think that this section adds much information that hasn't already been presented in the timeseries plots. I would suggest to either summarize the results in a meaningful way or to remove it. For example, can you draw any conclusions about the model's bias with regard to the different types of observations? Why are the biases with the J observations lower than the P observations, and why do Tables 3 and 4 show the contrary? The authors should either present a full analysis of the differences in the biases (i.e. instrument type, PBL height, time of day, concentration in the plume, location, windspeed and direction, etc), or simply report on the range of uncertainty that is found using these three datasets which is already quite useful information in terms of assessing the model's skill.

The Taylor diagram does not contain any information about biases, just correlations, normalized standard deviations, and centered/normalized root mean square errors. For this, we are hesitating to remove the Taylor diagram (see also comments by referee #2), because it allows to compare all different data sets regardless of their underlying biases.

However, as stated above, we revised our statistical analysis which now contains a bias corrections for the J and P datasets, but not for the C dataset, as we explain in the revised text.

We further guess that the referee here (in agreement with referee #2) refers to overall deviations (between bias-corrected model results and observations) and not the overall bias. Therefore, we added some more discussion and analysis to the revised text.

4.1 Theoretical Forecast Skill: I'm not convinced that the Taylor diagram brings any additional information that can't be deduced from Figure 12.

We agree, the Taylor diagram does not give any additional information. Neither does the respective explanation. We removed the whole sentence including the diagram.

4.2 Expected Skill Score: Please explain exactly what the expected skill score is because the fact that the model's skill does not decrease in the same manner as the theoretical skill score does not make sense to me. If we assume that the analysis is a "perfect simulation", and compared to the forecast simulation the theoretical forecast skill decreases to almost zero by day 6, how is it possible that the expected forecast skill in comparison to the observations is essentially the same on day 1 as day 6?

We came to the conclusion that the term "expected skill score" is badly chosen and highly misleading. This has also been pointed out by referee #2. Thus, we now use "actual skill score", which refers to the real observations. We give an explanation on different results of the two skills:
"Whereas the theoretical skill score is defined to measure the skill, averaged over the entire model domain, the actual skill score compares the model results to observational data. The latter measure the

downwind methane plumes, which are easier to forecast than the variability of the methane background in the overall model domain."

The authors present the model biases using different observations but more explanation or interpretation would be appreciated. For example, on page 19 line 1 it is stated that ". . .Sv is highest for J1 and J2. . ." but no explanation/speculation is offered as to why.

According to the different flight patterns between the J, C and the P observations, J "measures" the vertical gradient, P the small scale horizontal gradients, and C the larger scale horizontal gradients of the column integrated methane. Thus, the deviations of the model results seem to differ or to be inconsistent, which is however not the case. This is better explained in the revised text.

Again, I don't think that the Taylor diagram presented in Figure 16 adds any new information. It is clear from the plots in Figures 14 and 15, that the model's skill score for predicting the J observations is higher than for the other observations, especially the P observations. What would be interesting is for the authors to offer an explanation as to why this is the case. Why is there more variability in the skill score for the P observations than for the J or C observations? Unless the authors can draw some interesting conclusions such as this, I would suggest removing the Taylor diagrams and replacing them with the HALO and D-FDLR flight patterns.

We removed Figure 16 and the corresponding section.

Page 20, line 7: The authors state "All forecast days show a normalized standard deviation close to 1. . .meaning that all forecast days show similar amplitudes. . .". In theory, this can't be deduced from the standard deviation alone.

The referee is right! However, we removed this section in the revised manuscript, according to your comment above.

Technical Corrections: The author should go through the entire paper, especially (but not only) Section 3.1 and make sure they are consistent with either using the past or present tense.

We now use past tense in the revised manuscript. Section 3.1 still uses some past tense, as it refers to the sampling of the data in 2018.

Abstract: Line 4: change "measuring" to "measurement" Line 8: Change the sentence to read "In order to help with the flight planning during the campaigns. . ."

We changed both in the revised manuscript.

1 Introduction: page 2, Line 20: change "climate change strategies" to "climate change mitigation strategies"

We added the term "mitigation" to the sentence.

2 Evaluation of Analysis Simulation: Page 12, line 10: change sentence to ". . .observed peaks in the afternoon flight are lower those of the morning flight."

Page 13, line 4: change "very precisely" to "more precisely"
Page 13, line 6: change "constant offset" to "systematic bias"
Page 13, line 12: change ". . .simulated boundary layer. . ." to ". . .simulated boundary layer height. . ."

We corrected all points in the revised manuscript.

5 Discussion: page 22, line 5: There is something wrong with the sentence "This the intended result given the fact, . . .". Perhaps a word is missing.

We changed the sentence to: "This is the intended result considering that  . . .".

---

## Author Comment (AC2) · 26 Feb 2020

In black we repeat the referees comments, in red are our replies.

**Anonymous Referee #2**

This paper describes the setup and application of a nested atmospheric transport modeling system to support an aircraft measurement campaign with daily model forecasts of methane over a coal-mining region in Poland. The performance of the forecasts is assessed both in terms of theoretical skill (comparing 1-6 day CH4 forecasts with analyses) and in terms of "expected skill" (or actual skill) by comparison with aircraft observations of total column and in situ CH4. The model is shown to be capable of simulating the structures and amplitudes of the CH4 observations well, although this is more a qualitative than a quantitative statement since there is a large uncertainty in the underlying emission inventories and since there is no comparison with other modeling systems.

Dear Referee, thank you very much for the appreciation of our work.
Indeed there is not yet a comparison with other modeling systems, but one further study with the WRF-Stilt model is in preparation.

The paper is well written, clearly structured, and the analyses are detailed (sometimes too detailed) and sound. MECO(n) is an impressively flexible model system capable of online nesting multiple instances of a regional model (COSMO) in a global model (ECHAM). The paper presents a relevant application of the model, which takes full advantage of its nesting and online processing capabilities (e.g. sampling the model fields at each time step along aircraft trajectories).

We are very grateful for this positive valuation of the MECO(n) model system and its presented application.

Supporting aircraft measurement campaigns requires models with sufficient resolution (much better than the horizontal and vertical distance travelled during the flights), but it is not clear a priori, what resolution is really needed and whether very high resolution brings sufficient added value to justify the additional computational cost. By comparing the results of two different model instances with a resolution of 7 km and 2.8 km, respectively, the paper shows that the results (CH4 sampled along the aircraft tracks) are very comparable and that the higher resolution does not bring a great benefit, though some small-scale details were better resolved. An interesting but also surprising finding is that the model skill (evaluated against observations) did not depend clearly on forecast lead-time, i.e. a 3- or 4-day forecast performed equally well as a 2-day forecast. Unfortunately, there is little discussion of this result.

Thank you for that comment, we now discuss actual forecast skill in more detail, regarding also your comments below and the comment of referee #1.
Overall, I consider the publication acceptable with minor revisions, but I have a few main points and a number of small corrections/suggestions.

Main points:
- There are two methane tracers, PCH4 and CH4_FX, the first one representing emissions only from coal mining and the second all emissions (anthropogenic+natural) plus background CH4. The anthropogenic emissions in CH4_FX are based on EDGAR v4.2FT2010. The authors need to check how large emissions in the USCB region are in EDGAR in comparison with the total emissions of the COMET ED v1 inventory used for PCH4. To my understanding, fugitive emissions from solid fuels belong to category 1B1 (see IPCC 1996 reporting guide-lines), which is available as separate category in the EDGAR inventory. Without a comparison of these numbers, it is difficult to understand the results presented in Figure 8, which suggest and overestimation of the amplitude of CH4 enhancements for the tracer CH4_FX but an underestimation for the tracer PCH4. Furthermore, how do coal-mining emissions compare with other emissions e.g. from agriculture in this region (according to EDGAR?).

We moved this analysis from the discussion section into Sect. 3.2.2 and expanded it further, discussing CH4_FX and PCH4.
We compared total EDGAR v4.2FT2010 to CoMet ED v1. Therefore, we took every grid cell, for which we had information in CoMet ED v1, summed the emissions and compared them to the point source emissions in the respective grid cell. We further compared the 1B1 sector of EDGAR v4.2FT2010, which makes up more than 96% of the emissions. Agriculture plays a minor role, here. Only 11.18 kt/a of methane arise from EDGAR v4.2FT2010 sector 4 in the region (longitude: 18.3 °E to 19.4 °E, latitude: 49.9 °N to 50.4 °N).

- The simulation results are biased low because of a too low background. This should not be surprising considering that the simulation was initialized from a monthly climatological average of a period, when atmospheric CH4 was lower than in 2018. This bias is thus arbitrary and not of interest for the study (we are much more interested in the excursions from the background), but it dominates much of the statistics discussed and presented in the tables. I therefore suggest computing an overall bias (e.g. mean difference averaged over all flight sections measuring background) and subtract this constant offset from all simulation data, at least when computing the RMSE and NMBE statistics. In the current tables, the RMSE is of the order of 0.1 umol/mol, which is of a similar magnitude as the amplitude of the observed CH4 peaks, which would actually suggest a very poor model skill.

We revised our analysis and included a bias correction for the P and J observations. As explained in the text, for the C observations, this correction is, however, not applicable.

- The discussion on forecast skill is rather lengthy, especially the discussion of the Taylor diagrams. I found it useful to summarize the results of all model-observation comparisons in a Taylor diagram as shown in Figure 11, but I am much less convinced of the use of Figures 13 and 16 summarizing the 1- to 6-day forecast skills at a single location (Fig. 13) and for the aircraft measurements (Fig. 16). Much of the information is already conveyed by the other figures. The discussion of Figures 13 and 16 is lengthy and not providing much additional insight.

Thank you for this comment; we agree with that point and removed figures 13 and 16, as well as the corresponding text passage.

Furthermore, one should be very careful in the interpretation of the results presented in Fig. 11, since much of the findings are simply a consequence of the different flight patterns. The high correlations in

the HALO in situ measurements (J1, J2), for example, are primarily due to the large altitude changes on these flights probing a large vertical gradient in CH4. But also horizontal flight patterns may critically affect the results, depending on the complexity of the pattern, the overall distance flown, the time spent in sampling background versus polluted air, etc.

Fig. 11 is Fig. 8 in the revised manuscript.  We completely agree with the referee and now explain the relation to the flight patterns (and measurement technique – vertical column vs. in-situ) in the revised manuscript.

Minor points:
- Page 2, Line 28: Please explain what you mean by "internal"

"internal" simply refers to the fact that this inventory was compiled in preparation for this measurement campaign and it was not publicly available. We removed the "internal" from the revised text.

- P4, L4: You mention that the simulation data can optionally be interpolated vertically.
Was such vertical interpolation applied, or were the simulated fields only taken from the closest vertical layer?

Here, we sampled the vertical "curtain" along the horizontal flight tracks on-line on the original vertical model grids for direct output. This "curtain" was then further sub-sampled onto the flight altitude by linear interpolation (for P and J data) and by integration up to the flight altitude (for C data).
We mention this in the revised manuscript  on P10, L 1-2.

- P6: Figure 3 could be improved. The black text in the blue boxes is a bit difficult to read.

We chose a lighter color for the revision.

- P7, L4: Were the O1D and Cl fields obtained from a full chemistry simulation?

Yes. We added this information to the revised text.

- P7, Lines 13-18: I didn't really understand these sentences: Why do you need an "interpolation in time"? Why are data of time steps at 06:00 UTC and 12:00 UTC needed if the nudging requires two time steps AHEAD of the simulated time, which starts at 00:00 UTC?

This is solely due to technical constraints. We use the original nudging routines of the ECHAM5 base model. The nudging is applied in every time step, and the nudging fields (here the 6-hourly forecast data) are linearly interpolated in time. In addition, SST/SIC are consistently prescribed and also linearly interpolated in time in every model time step, however, in the standard configuration only every 12 hours. For this linear interpolation, the data are required 12 hours ahead.
We added this information to the revised text.

- P8, L9: I thing HPC stands for High Performance Computing (not "Performing").
Thanks. That was probably the spell checker ...

- P9, Figure 5: Why are the grey and green dashed lines with arrows going from right to left? This seems to suggest that e.g. the forecast starting at 12:00 is branched from an analysis, which has seen nudging data between 12:00 and 24:00 on that same day. Is this really true?

No. The issue is explained by the linear interpolation (see above) of the nudging data in time. A forecast starting at 12:00 UTC from the analysis simulation, requires the analysis simulation to be advanced until 12:00 UTC, which in turn (due to the time interpolation) required analysis nudging data until 24:00 UTC of that day. That is why we can branch off FC simulations with a lag of 12 hours, only. This is indicated by the dashed arrows and we actually do not have a better idea on how to visualize this.

- Figures 7 and 8: The flight patterns or at least the altitude profiles should also be shown in the main body of the paper, not just in the supplement, because this is essential information. It is important to know, for example, whether the individual peaks correspond to different plumes or whether the same plume was sampled back and forth multiple times. It is also important to know whether changes in CH4 mole fractions are due to changes in flight altitude rather than due to transecting a plume.

We now include figure 11, showing flight pattern and flight routes.

- Tables 2, 3 and 4: I suggest adding the correlation coefficients (or R-square as a measure of the variance explained).

We now added the correlation coefficient to all tables.

- P13, L21: I think it would be useful to show a vertical profile of CH4 for this flight to demonstrate that the model captures the vertical gradient of CH4 quite accurately.

We add Fig S1 to the supplement showing observations and model results as CH4 versus pressure altitude.

- P17, equations of skill scores: Does one of these skill scores correspond to the dashed line in Figure 11? Is so, please mention.

No. The dashed line refers to the centered normalized root mean square error (NRMSE).

- "Expected skill" doesn't sound right to me. What about "Actual skill", or "True skill"?

Thank you for this comment. Indeed, "expected skill" is not the most appropriate term. We changed it to "actual skill".

- P18, L1: For which period (how many days) did you compare the forecasts with the analysis simulation?

6 forecast days, starting each day between June 1 and June 22, 2018. Changed in text.
Corrections:
- Page 2, Line 28: "in the Upper Silesia" -> "in Upper Silesia"

- P4, L12: I suggest using "time step" instead of "time step length" here and in the following sentences.

We prefer to keep "time step length", which is the length of one time step in the model, whereas "time step" refers to the actual time step, e.g. the 1$^{st}$, 2$^{nd}$, 3$^{rd}$, etc.

- P13, L4: "very precisely". I would rather say "quite precisely"

- P13, L5: "at June" -> "on June"

- P13, L12: Change to "below the top of the boundary layer"

- P13, L24: "correlate well" has a positive connotation. "correlate closely" sounds better to me in this case.

- P14, L3: Change "Contrary," to either "In contrrast," or to "On the contrary," here and at other places.

- P14, L4: "expect that the model is able" -> "expect the model to be able" C5

- P15, L8: "suit well" -> "fit well"

Thank you for all the corrections, we changed them all (except one on P4, L12, see above) according to your specific suggestions.

- P15, L9: Isn't the NRMSE high rather than low?

You are right, this sentence is wrong. The correlation coefficient is low, but NRMSE is rather high. The sentence is changed now in the revised manuscript.

- P16, L1: "spacial" -> "spatial"

- P20, L14: "amplitude height" -> "amplitude"

- P22, L5: "This the intended result given" doesn't sound right.

- P24, L6: "the boundary layer is too low" -> "the boundary layer is too shallow". The top of the boundary layer can be too low, but not the boundary layer itself.

- P24, L13: "PCH4 correlates well with the observed methane emissions". There was no observation of emissions but only of concentrations.

- P24, L23: "might probably" -> "might"

- P24, L22: "forecast day" -> "forecast days"

All corrected.